# Adipocyte *microRNA-802* promotes adipose tissue inflammation and insulin resistance by modulating macrophages in obesity

**Yue Yang[1†], Bin Huang[1†], Yimeng Qin[1†], Danwei Wang[1†], Yinuo Jin[2], Linmin Su[1], Qingxin Wang[1], Yi Pan[1], Yanfeng Zhang[1], Yumeng Shen[3], Wenjun Hu[1], Zhengyu Cao[4]\*, Liang Jin[1]\*, Fangfang Zhang[1]\***

[1]State Key Laboratory of Natural Medicines, Jiangsu Key Laboratory of Druggability of Biopharmaceuticals, School of life Science and Technology, China Pharmaceutical University, Nanjing, China; [2]NanJing HanKai Academy, Nanjing, China; [3]State Key Laboratory of Natural Medicines, China Pharmaceutical University, Nanjing, China; [4]Jiangsu Key Laboratory of TCM Evaluation and Translational Research, School of Traditional Chinese Pharmacy, China Pharmaceutical University, Nanjing, China

**\*For correspondence:**
zycao1999@hotmail.com (ZC);
ljstemcell@cpu.edu.cn (LJ);
1620194592@cpu.edu.cn (FZ)

[†]These authors contributed equally to this work

**Competing interest:** The authors declare that no competing interests exist.

**Abstract** Adipose tissue inflammation is now considered to be a key process underlying metabolic diseases in obese individuals. However, it remains unclear how adipose inflammation is initiated and maintained or the mechanism by which inflammation develops. We found that *microRNA-802* (*Mir802*) expression in adipose tissue is progressively increased with the development of dietary obesity in obese mice and humans. The increasing trend of *Mir802* preceded the accumulation of macrophages. Adipose tissue-specific knockout of *Mir802* lowered macrophage infiltration and ameliorated systemic insulin resistance. Conversely, the specific overexpression of *Mir802* in adipose tissue aggravated adipose inflammation in mice fed a high-fat diet. Mechanistically, *Mir802* activates noncanonical and canonical NF-κB pathways by targeting its negative regulator, TRAF3. Next, NF-κB orchestrated the expression of chemokines and SREBP1, leading to strong recruitment and M1-like polarization of macrophages. Our findings indicate that *Mir802* endows adipose tissue with the ability to recruit and polarize macrophages, which underscores *Mir802* as an innovative and attractive candidate for miRNA-based immune therapy for adipose inflammation.

## Editor's evaluation

This important study utilizes a comprehensive array of animal and cellular models, alongside various techniques, to elucidate the mechanism by which adipose tissue miR-802 contributes to inflammation and metabolic dysfunction in obesity. The data are compelling, with consistent findings across replicates and different models. The work will be of interest to medical biologists working on obesity and metabolic diseases.

## Introduction

Obesity is a very powerful health determinant or indicator that facilitates the development and progression of several metabolic diseases, including insulin resistance and type 2 diabetes (*Ling and Rönn, 2019*; *Klein et al., 2022*). Adipose tissue is a highly dynamic metabolic organ that plays a central role in the regulation of energy homeostasis and controls glucose metabolism and insulin

sensitivity (*Scherer, 2006*; *Kang et al., 2008*). A hallmark of obesity is low-grade chronic inflammation in adipose tissue, characterized by the accumulation of macrophages and other immune cells, and by an increase in the levels of pro-inflammatory cytokines (*Pellegrinelli et al., 2022*; *Hägglöf et al., 2022*; *Kratz et al., 2014*). Persistent adipose tissue inflammation is now considered to have a pivotal role in obesity-associated insulin resistance (*Kohlgruber and Lynch, 2015*; *Burhans et al., 2018*). Resetting the immunological balance in obesity could represent an innovative approach for the management of insulin resistance and diabetes (*Brestoff et al., 2021*; *Lee et al., 2016*). However, the early triggers and signals that sustain adipose tissue inflammation in obesity remain elusive, limiting our ability to effectively intervene this growing public health issue.

Macrophages accumulate in the adipose tissue of obese mice and humans, where they form crown-like structures surrounding dying or dead adipocytes and are key contributors to inflammation and obesity-induced insulin resistance (*Weisberg et al., 2003*; *Hotamisligil, 2006*). The number of adipose tissue macrophages is tightly linked to the degree of insulin resistance and metabolic dysregulation (*Xu et al., 2003*; *Chawla et al., 2011*). Ablation of pro-inflammatory adipose tissue macrophages leads to a rapid improvement in insulin sensitivity and glucose tolerance, associated with marked decreases in local and systemic inflammation in obese mice (*Patsouris et al., 2008*; *Nomiyama et al., 2007*). Targeting the major inflammatory pathways is sufficient to counteract obesity-related systemic inflammation and insulin resistance (*Arkan et al., 2005*; *Patra et al., 2023*). However, the molecular links between lipid-overloaded adipocytes and inflammatory macrophages in obese adipose tissue remain elusive.

MicroRNAs (miRNAs) are small non-coding RNAs that post-transcriptionally regulate gene expression by binding to specific regions of target genes to prevent translation or promote mRNA degradation (*Ambros, 2004*). Emerging evidence suggests that miRNAs are key regulators in a variety of important metabolic organs and substantial contributors to the pathogenesis of complex diseases, including obesity-associated metabolic diseases (*Dumortier et al., 2013*; *Krützfeldt and Stoffel, 2006*). In the adipose tissue, miRNAs have dramatic effects on regulating the pathways that control a range of processes including lipogenesis, inflammation, and insulin signaling (*Arner and Kulyté, 2015*; *Thomou et al., 2017*). Moreover, mice with alterations in the levels of miRNAs in adipocytes show significantly enhanced inflammation and insulin resistance after feeding with a high-fat diet (HFD), further confirming the contribution of miRNAs to obesity-induced phenotypes (*Agbu and Carthew, 2021*; *Koh et al., 2018*). Therefore, adipose-derived miRNAs hold great promise for understanding adipose tissue dysfunction and the relationship between chronic inflammation and obesity and insulin resistance.

In this study, we demonstrated that *microRNA-802* (*Mir802*) promotes inter-cellular communication between lipid-overloaded adipocytes and macrophages, ultimately leading to adipose tissue inflammation and insulin resistance. Adipocyte *Mir802* levels are positively associated with obesity in mice and humans. Adipose tissue-specific overexpression of *Mir802* in mice fed an HFD exhibited increased severity of systemic insulin resistance compared with wild-type (WT) mice, which was accompanied by macrophage infiltration and a marked increase in adipose tissue inflammation. Adipose tissue-specific knockout of *Mir802* achieved the opposite result. Co-culture and other in vitro experiments revealed a vicious cycle of interactions between macrophages and adipocytes ectopically expressing *Mir802*. We established that *Mir802* expression is an inflammatory signal in adipocytes, and this effect occurs through sensitization of the NF-κB signaling pathway. Altogether, our data raise the possibility that manipulation of this microRNA action axis has therapeutic potential for treating adipose inflammation.

## Results

### *Mir802* elevation precedes macrophage accumulation

Consistent with previous studies from our and other laboratories (*Zhang et al., 2020*; *Kornfeld et al., 2013*), adipose from obese mice showed significantly higher *Mir802* expression than those from normal mice. To evaluate whether *Mir802* is involved in adipose inflammation and insulin resistance. We examined the expression profile of *Mir802*. We observed that *Mir802* progressively increased in adipose tissue from week 4 with the development of obesity in mouse models of genetic and dietary obesity (*Figure 1A, B*, *Figure 1—figure supplement 1A*). We next compared the expression of *Mir802* in different adipose depots and found that it was the highest in epididymal white

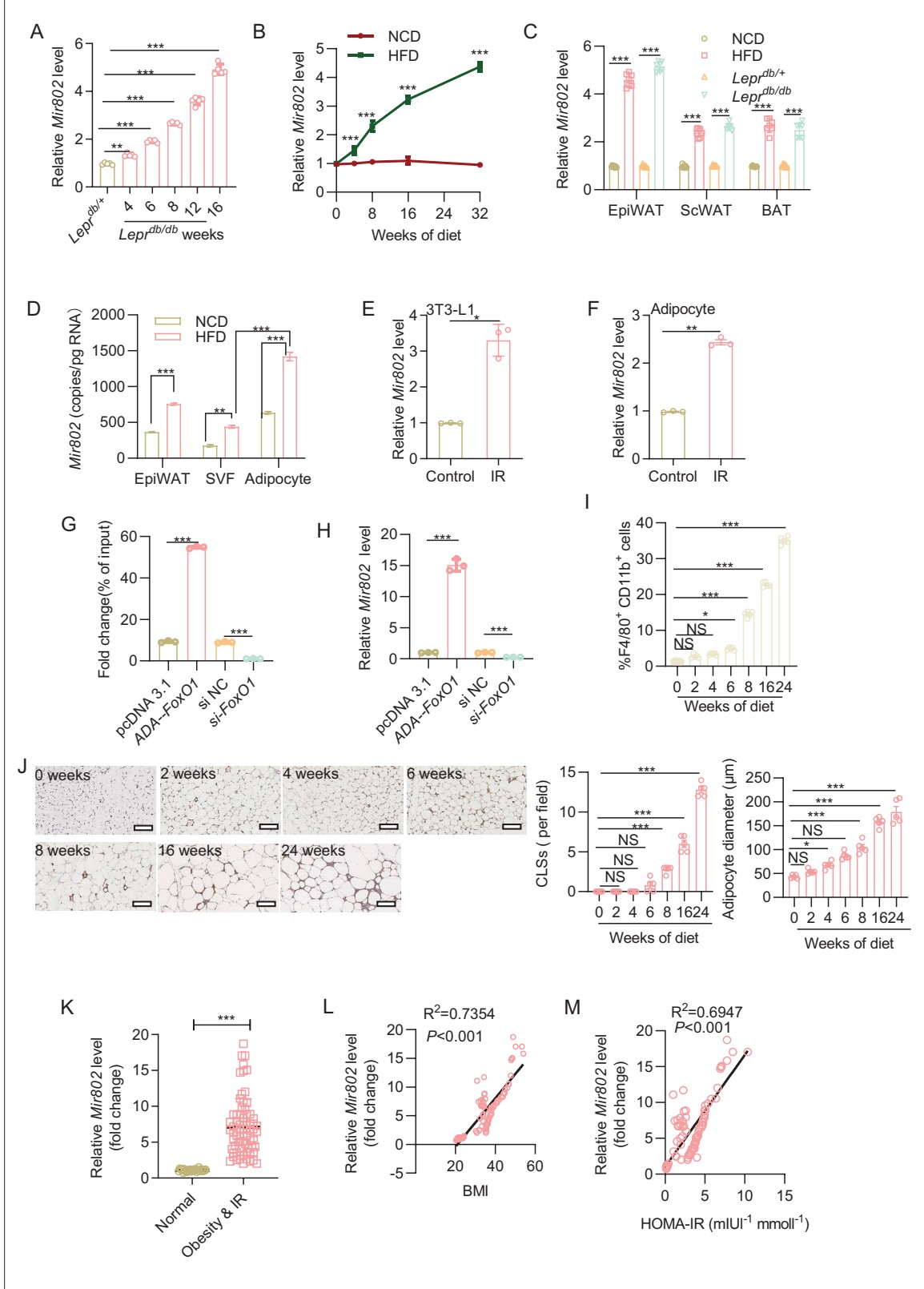

**Figure 1.** Obesity induced *Mir802* elevation precedes macrophage accumulation. (**A**) mRNA abundance of *Mir802* in the epiWAT of *Lepr*[db/db] or control mice at 4, 6, 8, 12, and 16 weeks (n=5). (**B**) mRNA abundance of *Mir802* in the epiWAT of mice fed a normal chow diet (NCD) or HFD for 0, 4, 8, 16, 24, and 32 weeks (n=5). (**C**) The expression level of *Mir802* in epiWAT, scWAT and BAT isolated from mice on HFD for 16 weeks or 10 weeks *Lepr*[db/db] mice (n=7). (**D**) Copy number of *Mir802* in mature adipocytes and stromal vascular fraction (SVF) of epiWAT isolated from mice on NCD or HFD for

*Figure 1 continued on next page*

*Figure 1 continued*

16 weeks (n=5). (**E–F**) *Mir802* expression levels in insulin resistance 3T3-L1 cell models (**E**) and insulin resistance WAT SVF cells models (**F**). (**G**) ChIP assays was performed to test the binding ability between *FoxO1* and *Mir802* promoter. (**H**) *Mir802* expression levels in the 3T3-L1 cells transfected with *ADA-FoxO1* or *FoxO1* siRNA. (**I**) F4/80 and CD11b positive cells in SVFs isolated from the epiWAT of mice fed an HFD for 0, 2, 4, 6, 8, 16, and 24 weeks (n=5). (**J**) Representative images of F4/80 staining (left) and quantification of crown-like structures (CLSs; middle) and adipose diameter (Right) in the epiWAT of mice fed an HFD (n=5). (**K**) Expression levels of *Mir802* in human subcutaneous adipose tissue ($n_{normal}$ = 25, $n_{obesity \& IR}$=70). Scatter plots of *Mir802* expression versus BMI (**L**) and HOMA-IR (**M**). Pearson's correlation coefficients (**r**) are shown. The fold of *Mir802* was calculated using $2^{-\Delta\Delta Ct}$. Data represent mean ± SEM. p-values obtained using a two-tailed unpaired Student's *t*-test (**E, F, K**) or two-way ANOVA (**A–D, G–J**) are indicated. *p<0.05, **p<0.01, ***p<0.001. Relative levels of *Mir802* were normalized to *U6*. epiWAT: epididymal white adipose tissue, scWAT: subcutaneous white adipose tissue, BAT: brown adipose tissue.

The online version of this article includes the following figure supplement(s) for figure 1:

**Figure supplement 1.** Obesity induced Mir802 elevation precedes macrophage accumulation.

adipose tissue (epiWAT; *Figure 1C*). We further isolated mature adipose tissue and stromal vascular fraction (SVF) from epiWAT to examine the expression of *Mir802*. We found that *Mir802* expression was substantially higher in mature adipocytes than in SVF in both mice fed a normal chow diet (NCD) and those fed an HFD (*Figure 1D*). Through in vitro experiments, we found that *Mir802* was dramatically increased in the insulin resistance cell models (*Figure 1E, F*, *Figure 1—figure supplement 1C, D*). We have determined that the upregulation of islet *Mir802* during obesity is mediated by *Forkhead box O1* (*FoxO1*; *Zhang et al., 2020*). *FoxO1* is predominantly expressed in WAT and is upregulated in WAT of the obese mice (*Teaney and Cyr, 2023*). Here, we found that *ADA-FoxO1* exhibited significantly the higher binding ability of *FoxO1* to *Mir802* promoter compared to control via ChIP assay (*Figure 1G*) and over-expression *FoxO1* upregulated *Mir802* expression (*Figure 1H*). These findings suggest that obesity induced *Mir802* expression via *FoxO1* and upregulation of *Mir802* in adipocytes may be functionally involved in the pathogenesis of obesity-associated disorders.

Initial studies have indicated that macrophages are responsible for most inflammatory events in adipose tissue (*Weisberg et al., 2003*; *Hotamisligil, 2006*). However, what initiates macrophage infiltration or the resultant inflammatory cascade is still not well defined. We hypothesized that the elevation of *Mir802* in adipocytes is associated with adipose inflammation and insulin resistance. To evaluate this hypothesis, we analyzed the relationship between *Mir802* elevation and macrophage infiltration during the progression of diet-induced obesity (DIO). We first carried out a set of flow cytometric analyses to determine the dynamic alterations of macrophages in collagenase-digested SVF from epiWAT. Starting from 8 week, there was a gradual and continuous increase in the number of CD11b/F4/80 double-positive macrophages observed in obese mice (*Figure 1I*, *Figure 1—figure supplement 1E*). Immunohistochemical analysis of F4/80 expression also revealed that the number of macrophages continued to increase in the epididymal fat pads of obese mice as compared to that in mice on a normal diet (*Figure 1J*). The dynamic increase in *Mir802* preceded the infiltration of macrophages, indicating that *Mir802* may play a critical role in the occurrence of adipose inflammation.

To gain additional insight into the clinical importance of *Mir802* in obese fat, we analyzed the expression of *Mir802* in samples of human subcutaneous adipose tissue. Levels of *Mir802* expression were significantly higher in obese subjects (body mass index [BMI]=38.30 ± 5.82 kg/m², fasting plasma glucose = 8.39 ± 1.54 mM, homeostatic model assessment for insulin resistance [HOMA-IR]=3.77 ± 1.97) than in lean ones (BMI = 22.09 ± 1.09 kg/m², fasting plasma glucose = 4.84 ± 0.53 mM, HOMA-IR=0.21 ± 0.06; *Figure 1K*, *Figure 1—figure supplement 1F*). Pearson's correlation analysis showed that the BMI and HOMA-IR were positively associated with *Mir802* abundance in subcutaneous fat (*Figure 1L and M*). The same phenomenon was also observed in RNA-FISH analysis (*Figure 1—figure supplement 1G*), indicating that upregulation of *Mir802* in the adipose tissue during obesity is conserved in humans.

## Adipose-selective overexpression of *Mir802* aggravates inflammatory cascade in obese mice

To further assess the role of adipocyte *Mir802*, we generated adipose-selective *Mir802* konck-in (*Mir802* KI) mice by crossing *Mir802*^ki/ki^ mice (*Zhang et al., 2020*) with animals expressing Cre recombinase under the control of the promoter of *Adiponectin* (*Figure 2—figure supplement 1A*, B). Real-time PCR analysis confirmed that the overexpression of *Mir802* was restricted in the adipose tissues

of the *Mir802* KI mice, *Mir802* expressions was up-regulated about 150 times, whereas its expression in other organs was not affected except for BAT (*Figure 2—figure supplement 1C*), and the upregulation of *Mir802* was limited to adipocytes and was not observed in SVFs (*Figure 2—figure supplement 1D*). There was no obvious difference in food intake, body weight, glucose content, and adiposity between *Mir802* KI mice and their WT littermates in both male and female when they were fed with NCD (*Figure 2—figure supplement 1E–H*). We then fed the mice an HFD and performed metabolic and histological analyses. We detected the presence of adipose inflammation, typified by macrophage crown-like structures (CLSs) in epiWAT at 8 weeks in *Mir802* KI mice, which was earlier than their WT littermates, and the number of CLSs was almost doubled at 16 weeks (*Figure 2A*). No significant differences were observed in CLSs between the control and *Mir802* KI mouse groups treated with normal chow diet (NCD, *Figure 2—figure supplement 1I*). Consistently, flow cytometric analysis showed that HFD-induced elevation in the number of CD11b$^+$F4/80$^+$ macrophages in the SVF of epiWAT in adipose-specific *Mir802* KI mice was significantly higher than that in WT littermates in both male and female (*Figure 2B*, *Figure 2—figure supplement 1J*). In *Mir802* KI mice fed on HFD for 16 weeks, the number of classically activated proinflammatory M1 macrophages (defined as CD86$^+$CD206$^-$) was significantly higher than that of alternatively activated anti-inflammatory M2 macrophages (defined as CD86$^-$CD206$^+$) in epiWAT (*Figure 2C*, *Figure 2—figure supplement 1K*). In line with this finding, epiWAT from dietary-obese *Mir802* KI mice exhibited obviously higher mRNA expression of the M1 macrophage–related genes (*Ccl2, Il1b, Il6, Tnfa, Inos,* and *Ifng*) but significant reductions of M2 macrophage–related genes (*Il10, Chil3, Arg1,* and *Fizz1*; *Figure 2D*). Similarly, HFD also increased the level of several inflammatory factors (chemokine ligand 2 [CCL2], interleukin IL-6, IL-1β, and tumor necrosis factor TNF-α) in the serum of *Mir802* KI mice (*Figure 2E–H*).

We next explored whether the aggravation of adipose inflammation in adipose-selective *Mir802* KI mice in both male and female were associated with exacerbation of metabolism and insulin sensitivity. We found that in *Mir802* KI mice, HFD induced weight gain (*Figure 2I*, *Figure 2—figure supplement 2A*) and hyperglycemia (*Figure 2J*) both in male and female. HFD also induced adiposity in *Mir802* KI mice, which mainly manifested in the expansion of visceral WAT (*Figure 2K and L*). MRI analysis confirmed that HFD induced an increase in visceral WAT in *Mir802* KI mice (*Figure 2M*). We next monitored the dynamic changes in insulin sensitivity at different time points (0, 4, 8, 16, and 30 weeks) after feeding the two groups of mice with an HFD. As expected, *Mir802* KI mice on a HFD exhibited progressive development of glucose intolerance (*Figure 2N*, *Figure 2—figure supplement 2B-F*) and insulin resistance (*Figure 2O*, *Figure 2—figure supplement 2G-K*) at 8 weeks, as compared to their WT littermates. These differences became even more obvious after 16 and 30 weeks, coupled with an increase in fasting insulin levels (*Figure 2P*) and HOMA-IR (*Figure 2Q*). Collectively, these effects of adipose-selective overexpression of *Mir802* show that *Mir802* is sufficient for the recruitment of macrophages into obese adipose tissue and for the initiation and propagation of the inflammatory cascade.

## *Mir802* depletion ameliorates obesity-induced metabolic dysfunction

Given the striking effects of adipose-selective overexpression of *Mir802* on metabolism, we next investigated whether selectively ablated *Mir802* in adipose tissue could mitigate metabolic disturbance and inflammation induced by obesity. We generated *Mir802* conditional knockout mice using the Cre/Lox system (*Figure 3—figure supplement 1A*). *Mir802*$^{fl/fl}$ were crossed with *Adipoq*-Cre transgenic animals to selectively ablate *Mir802* in adipose tissues (*Figure 3—figure supplement 1B*). Expression analysis showed that total *Mir802* levels were reduced by approximately 70% in the adipose tissue but not in SVFs of *Mir802* KO mice compared with WT littermates (*Figure 3—figure supplement 1C, D*). The knockout of *Mir802* in adipose tissue did not alter food intake, body weight, glucose levels, and adiposity compared with their WT littermates in both males and females when they were fed with NCD (*Figure 3—figure supplement 1E–I*); however, this approach could prevent HFD-induced weight gain and hyperglycemia (*Figure 3A, B*, *Figure 3—figure supplement 1J*). Adipose-selective ablation of *Mir802* also alleviated HFD-induced adiposity, mainly by reducing the expansion of visceral WAT, including epiWAT and retro-peritoneal WAT (*Figure 3C and D*). MRI analysis confirmed this result (*Figure 3E*). Histological and FACS analysis showed that *Mir802* depletion reduced macrophage infiltration, which mainly manifested as a decrease in the number of CLSs and macrophages (*Figure 3F, G*, *Figure 3—figure supplement 1K*), while there was minimal difference observed between the two

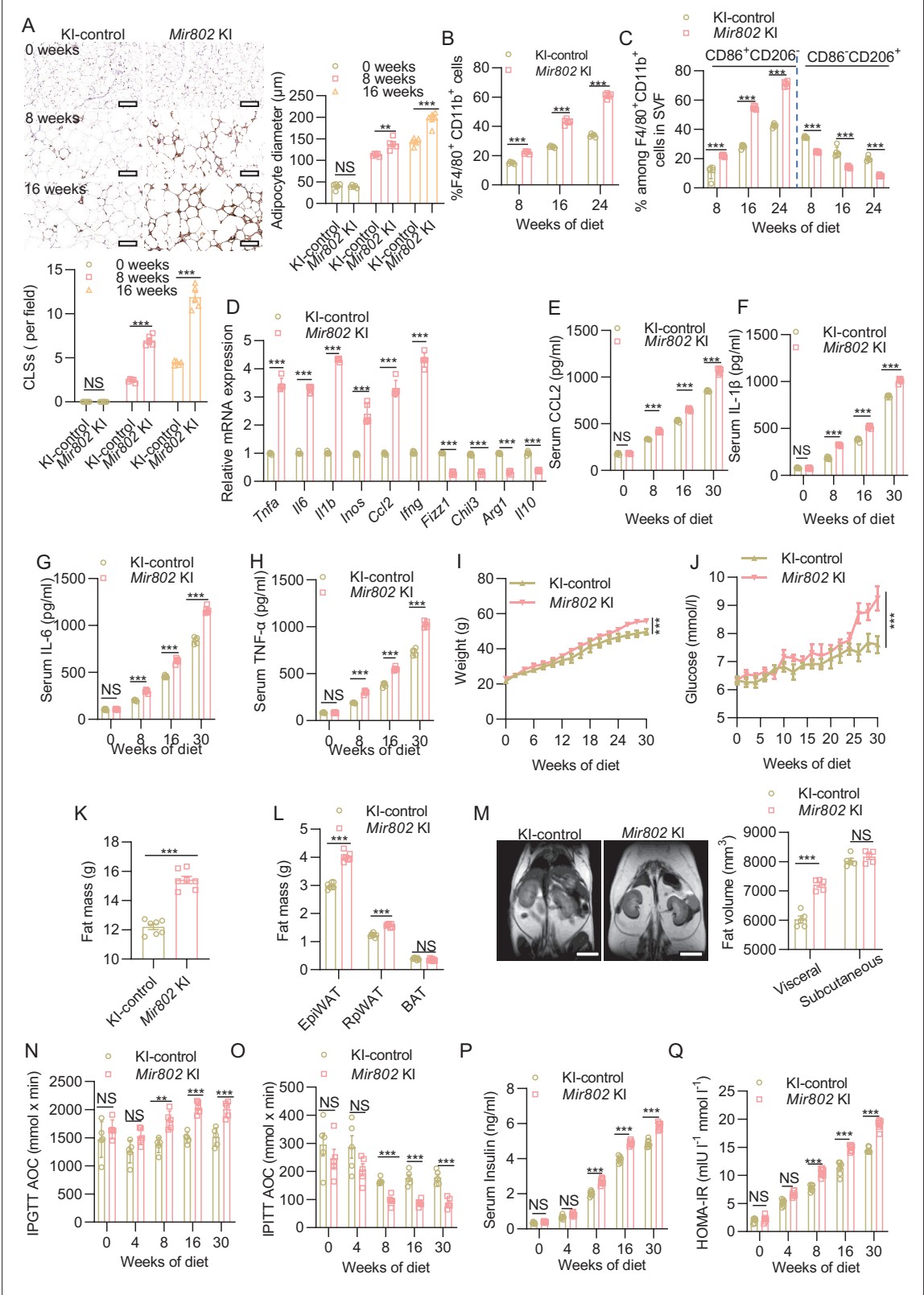

**Figure 2.** Adipose tissue-specific overexpression of *Mir802* exacerbates adipose tissue inflammation and leads to metabolic dysfunction.
(**A**) Representative images of F4/80 staining (top), quantification of CLSs (bottom) and adipose diameter (right) in epiWAT of WT or *Mir802* KI mice on HFD for 0, 8, and 16 weeks (n=5). Scale bar: 40 µm. (**B**) Percentage of F4/80+/CD11b+ total macrophages in the epiWAT of *Mir802* KI and KI-control mice fed with HFD (n=5). (**C**) M1 (CD86+CD206−) and M2 (CD206+CD86−) within the macrophage population (n=5). (**D**) qRT-PCR analysis for mRNA levels

*Figure 2 continued on next page*

*Figure 2 continued*

of the M1 and M2 markers in the epiWAT of mice on KI-control or *Mir802* KI at 16 weeks (n=5). (**E–H**) Serum levels of CCL2 (**E**), IL-1β (**F**), IL-6 (**G**), and TNF-α (**H**) of *Mir802* KI and control mice fed with HFD for 0, 8, 16, and 30 weeks (n=5). (**I, J**) Dynamic changes in body weight (**I**) and glucose (**J**) in WT and *Mir802* KI mice during 30 weeks of HFD feeding (n=5). (**K, L**) Fat mass of whole body (**K**) and individual tissues (**L**) (n=7). (**M**) Representative coronal section MRI images and visceral and subcutaneous adipose tissue volume of HFD-fed control and *Mir802* KI mice (n=5). (**N, O**) Area over the curve (AOC) of the blood glucose level was calculated via intraperitoneal glucose tolerance tests (IPGTTs, 2 g/kg, N, n=5) or intraperitoneal insulin tolerance tests (IPITTs; 0.75 U/kg, **O**, n=5). (**P**) Fasting insulin (FINS) levels of HFD-fed mice were measured by ELISA (n=7). (**Q**) HOMA-IR was calculated with the equation FBG (mmol l$^{-1}$)×FINS (mIU l$^{-1}$)/22.5. Data represent mean ± SEM. Differences between groups were determined by ANOVA (**A–J**, **L**, and **N–Q**) or two-tailed unpaired Student's *t*-test (**K**). *p<0.05, ***p<0.001. Gene levels were normalized to *Rn18s* abundance.

The online version of this article includes the following source data and figure supplement(s) for figure 2:

**Figure supplement 1.** Adipose tissue-specific overexpression of Mir802 exacerbates adipose tissue inflammation.

**Figure supplement 1—source data 1.** Related to *Figure 2—figure supplement 1B*.

**Figure supplement 2.** Adipose tissue-specific overexpression of Mir802 leads to metabolic dysfunction.

groups fed with NCD (*Figure 3—figure supplement 1L*). Notably, the *Mir802* KO mice exhibited obvious reductions in mRNA expression of the M1 macrophage-related genes (*Ccl2*, *Il1b*, *Il6*, *Tnfa*, *Inos*, and *Ifng*) but significant upregulation of M2 macrophage-related genes (*Fizz1*, *Chil3*, *Arg1*, and *Il10*; *Figure 3H*). The *Mir802* KO mice also markedly blunted HFD-induced elevation in serum levels of several inflammatory factors (TNF-α, IL-6, IL-1β, and CCL2; *Figure 3I*). In addition, the insulin resistance and glucose intolerance induced by an HFD were ameliorated by *Mir802* depletion (*Figure 3J, R*, *Figure 3—figure supplement 1M-R*). These phenomena were the same both in male and female *Mir802* KO mice.

We next examined the activity of *Mir802* in obese adipose tissues in which inflammation had already been established. We performed the acute deletion of adipocyte *Mir802* that did not influence whole-body weight. To address this question, we depleted *Mir802* in eWAT using an approach of adeno-associated virus (AAV, *Mir802* eWAT KD) to 16-week-old DIO mice that had been fed an HFD since they were 4 weeks old (*Figure 3—figure supplement 2A*). After 7 days, we detected 70% lower expression of *Mir802* compared with the control in the epididymal fat pad; *Mir802* expression was unaffected in other organs except for BAT (*Figure 3—figure supplement 2B*). The weight and number of CLSs were lowered with *Mir802* sponge treatment (*Figure 3—figure supplement 2C*, *Figure 3L*), and the reduction in macrophage infiltration was confirmed by CD11b and F4/80 flow cytometry analysis (*Figure 3M*, *Figure 3—figure supplement 2D*). Phenotypic analysis indicated that *Mir802* inhibitor also lowered the M1 (CD86⁺CD206⁻) macrophage fraction, while it increased the M2 macrophage (CD206⁺CD86⁻) fraction (*Figure 3N*, *Figure 3—figure supplement 2E*). DIO led to upregulated mRNA expression of proinflammatory cytokines (*Il1b*, *Il6*, and *Tnfa*) in the adipose tissue that was suppressed in the *Mir802* eWAT KD mice (*Figure 3O*). *Mir802* inhibitor treatment also ameliorated insulin resistance and glucose intolerance in DIO mice (*Figure 3P*, *Figure 3—figure supplement 2F*). These results clearly show that *Mir802* inhibitor treatment suppresses preexisting adipose inflammation, which strongly suggests that *Mir802* is required for the maintenance of inflammatory reactions in obese adipose tissue.

## Interplay between *Mir802* ectopically expressed adipocytes and macrophages

We next analyzed the cellular interplay via which inflammation develops in obese adipose tissue. Based on the findings of the in vivo experiments summarized above, we hypothesized that obese adipose tissue upregulates *Mir802*, and *Mir802*-overexpressing adipocytes in turn recruit and activate macrophages. To test this hypothesis, we first co-cultured bone-marrow-derived differentiated macrophages (BMDMs) with differentiated WAT SVF cells obtained from either lean or obese mice to assess whether adipose tissue from obese mice can influence the behavior of BMDMs (*Figure 4A*). EdU assays and flow cytometric analysis showed that obese differentiated WAT SVF induced the proliferation of BMDMs, whereas lean fat did so only mildly (*Figure 4—figure supplement 1A, B*). Transwell co-culture further showed that obese WAT differentiated SVF also promoted BMDMs migration and invasion (*Figure 4B*). We next explored the effects of obese mice' WAT differentiated SVF on the characteristics of BMDMs. After co-culture BMDMs and differentiated WAT SVF of obese mice, BMDMs

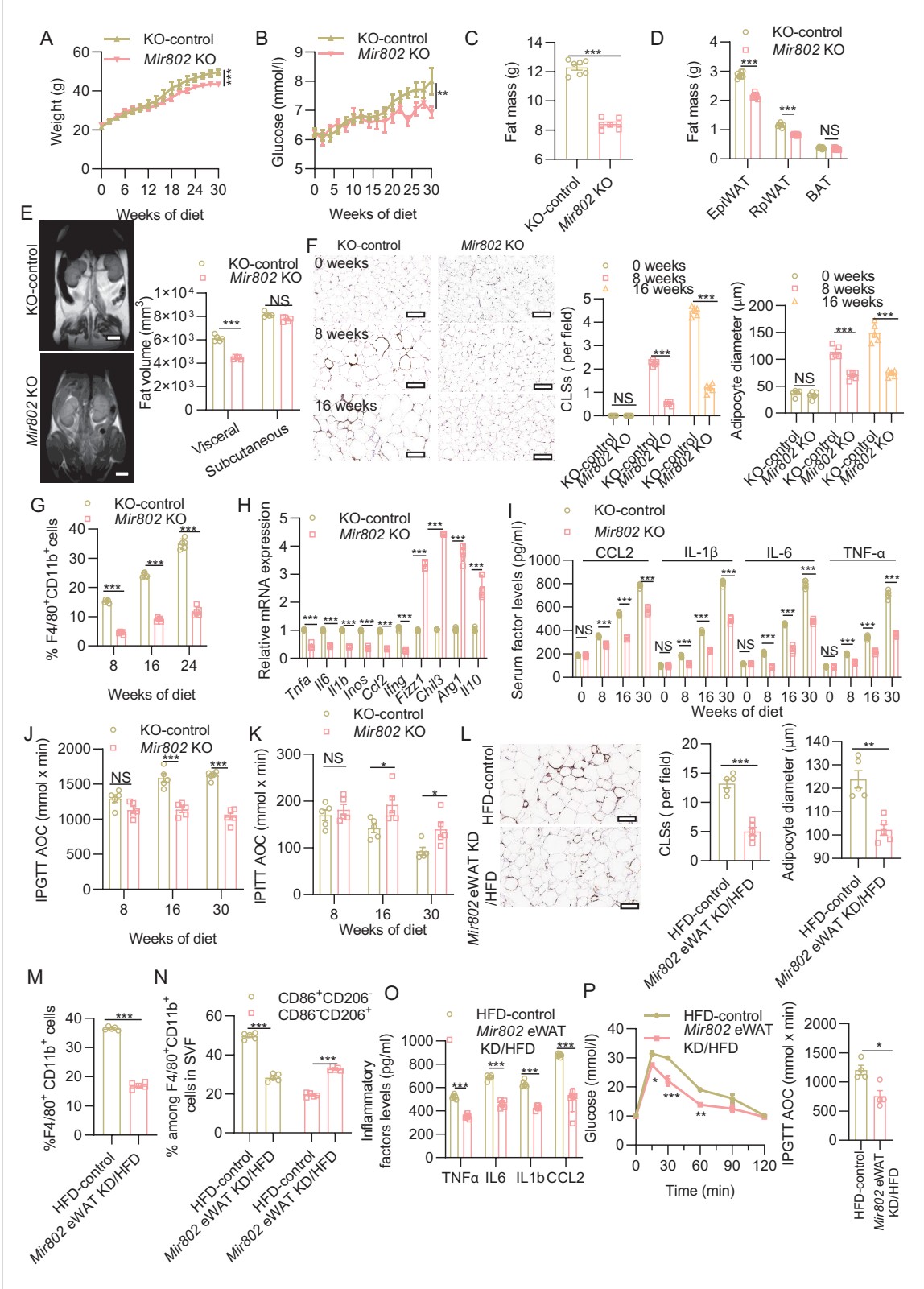

**Figure 3.** Adipose tissue–specific ablation of *Mir802* protects mice from obesity-induced metabolic dysfunction. (**A–B**) Dynamic changes in body weight (**A**) and glucose (**B**) of KO control and *Mir802* KO mice during 30 weeks of HFD feeding (n=7). (**C–D**) Fat mass of whole body (**C**) and individual tissues (**D**) (n=7). (**E**) Representative coronal section MRI images and visceral and subcutaneous adipose tissue volume of HFD-fed control and *Mir802* KO mice (n=5). (**F**) Representative images of F4/80 staining (left), quantification of CLSs (middle) and adipose diameter (right) in epiWAT of WT or *Mir802*

*Figure 3 continued on next page*

*Figure 3 continued*

KO mice on HFD for 0, 8, and 16 weeks (n=5). Scale bar: 40 µm. (**G**) Cells isolated from SVFs of epiWAT in *Mir802* KO and WT mice fed with HFD for 8, 16, and 24 weeks were subjected to flow cytometry analysis for percentage of CD11b⁺/F4/80⁺ total macrophages (n=5). (**H**) qRT-PCR analysis for the mRNA levels of the M1 and M2 markers in epiWAT of mice on HFD 16 weeks (n=5). (**I**) Serum levels of CCL2, IL-1β, IL-6, TNF-α determined with ELISA (n=5). (**J–K**) AOC of the blood glucose level was calculated via IPGTT (1.5 g/kg, **J**, n=5) or IPITT (0.75 U/kg, **K**, n=5). (**L**) Representative images of F4/80 staining (left), quantification of CLSs (middle) and adipose diameter (right) (n=5) in the epiWAT of WT or *Mir802* KO mice. Scale bar: 40 µm. (**M–N**) The percentage of CD11b⁺/F4/80⁺ total macrophages (M, n=5) and M1 (CD86⁺CD206⁻), and M2 (CD206⁺CD86⁻) within the macrophage population (N, n=5) in the SVFs isolated from epiWAT in the HFD-control or *Mir802* eWAT KD/HFD mice. (**O**) Serum levels of TNF-α, IL-6, IL-1β, CCL2 determined with ELISA (n=6). (**P**) IPGTT was performed in HFD-control mice or *Mir802* eWAT KD/HFD mice(n=5). Data represent mean ± SEM. Differences between groups were determined by ANOVA (**A–B, D, E–K, N–P**) or two-tailed unpaired Student's *t* test (**C, L–M**). ***p<0.001. Gene levels were normalized to *Rn18s* abundance.

The online version of this article includes the following source data and figure supplement(s) for figure 3:

**Figure supplement 1.** Adipose tissue–specific ablation of Mir802 protects mice from obesity-induced metabolic dysfunction.

**Figure supplement 1—source data 1.** Related to *Figure 3—figure supplement 1B*.

**Figure supplement 2.** Knockdown Mir802 in the eWAT of HFD mice restores adipose function.

had elevated expression of classical activation (M1-like) marker CD86, whereas the alternative activation marker (M2-like) CD206 was decreased (*Figure 4C*, *Figure 4—figure supplement 1C*). The results of ELISA indicated that obese mice' WAT SVF-induced BMDMs were predominantly polarized to pro-inflammatory macrophages (*Figure 4D*).

We next to investigate which factors induce the crosstalk between adipose and macrophage. We plated BMDMs in boyden chambers and treated them with a medium conditioned with obese mice' WAT differentiated SVF or lean mice' WAT differentiated SVF, the number of BMDMs that migrated through the pores between chamber wells with obese mice' WAT differentiated SVF conditioned medium was significantly higher than the number of cells cultured in lean mice' WAT differentiated SVF conditioned medium (*Figure 4E*). qRT-PCR results showed that *Mir802* expression in the BMDMs has no change (*Figure 4F*), while ELISA results showed that conditioned medium of obese differentiated WAT SVF can secrete more humoral factors known to induce BMDMs migration, especially CCL2 (*Figure 4G*).

To further confirm the function of *Mir802* in adipose tissue, the adipocyte cell line 3T3-L1 was transfected with *Mir802* mimics (*Mir802*) or *Mir802* inhibitor (anti-*Mir802*). We then explored the effect of *Mir802* ectopically expressed 3T3-L1 cells on the macrophage cell line RAW 264.7 in co-culture (*Figure 4—figure supplement 1D*). The knockdown and overexpression efficiencies were approximately 80% and 240-fold, respectively (*Figure 4—figure supplement 1E*). First, we found that the *Mir802* levels were no different in the RAW 264.7 cells (*Figure 4H*), *Mir802*-overexpressing 3T3-L1 cells had no effect on the proliferation and lipid droplet production of RAW 264.7 cells (*Figure 4—figure supplement 1F, G*). However, *Mir802*-overexpressing 3T3-L1 cells promoted the migration and invasion of RAW 264.7 cells, whereas 3T3-L1 cells knocked down by anti-*Mir802* had the opposite effect (*Figure 4I*). *Mir802* mimics-transfected 3T3-L1 cells also promoted RAW 264.7 cells M1-like polarization (*Figure 4J*, *Figure 4—figure supplement 1I*). We also found higher level of CCL2 in the medium conditioned with *Mir802*-overexpressed 3T3-L1 cells (*Figure 4—figure supplement 2A*). Moreover, we found that without co-culture, only CCL2 was increased, TNF-α, IL-6, IL-1β levels have no difference (*Figure 4—figure supplement 2B*). Additionally, we have performed the migration/invasion assay with no adipocyte, co-culture with adipocyte, co-culture with adipocyte transfected *Mir802* mimics, co-culture with adipocyte transfected *Mir802* mimics and added emapticap pegol (also known as NOX-E36, CCL2 inhibitor). The results showed that no adipocyte, RAW 264.7 cells almost have no ability to migration and invasion, co-culture with *Mir802* mimics promoted the migration and invasion of RAW 264.7 cells compared to co-culture with miR-NC, while blocking CCL2 in *Mir802*-overexpressed 3T3-L1 cells exhibited reduced RAW 264.7 cells recruitment ability (*Figure 4—figure supplement 2C*).

We next investigated whether the metabolic phenotypes in adipose-selective *Mir802* KI mice depend on the presence of macrophages. To this end, both adipose-specific *Mir802* KI mice and WT littermates were intraperitoneally injected with clodronate-conjugated liposomes (CLOD-liposomes) to deplete macrophages (*Hui et al., 2015*), and PBS-liposomes as vehicle control. The clodronate liposomes treatment ameliorated systematic inflammation, as displayed by the decreased levels of serum

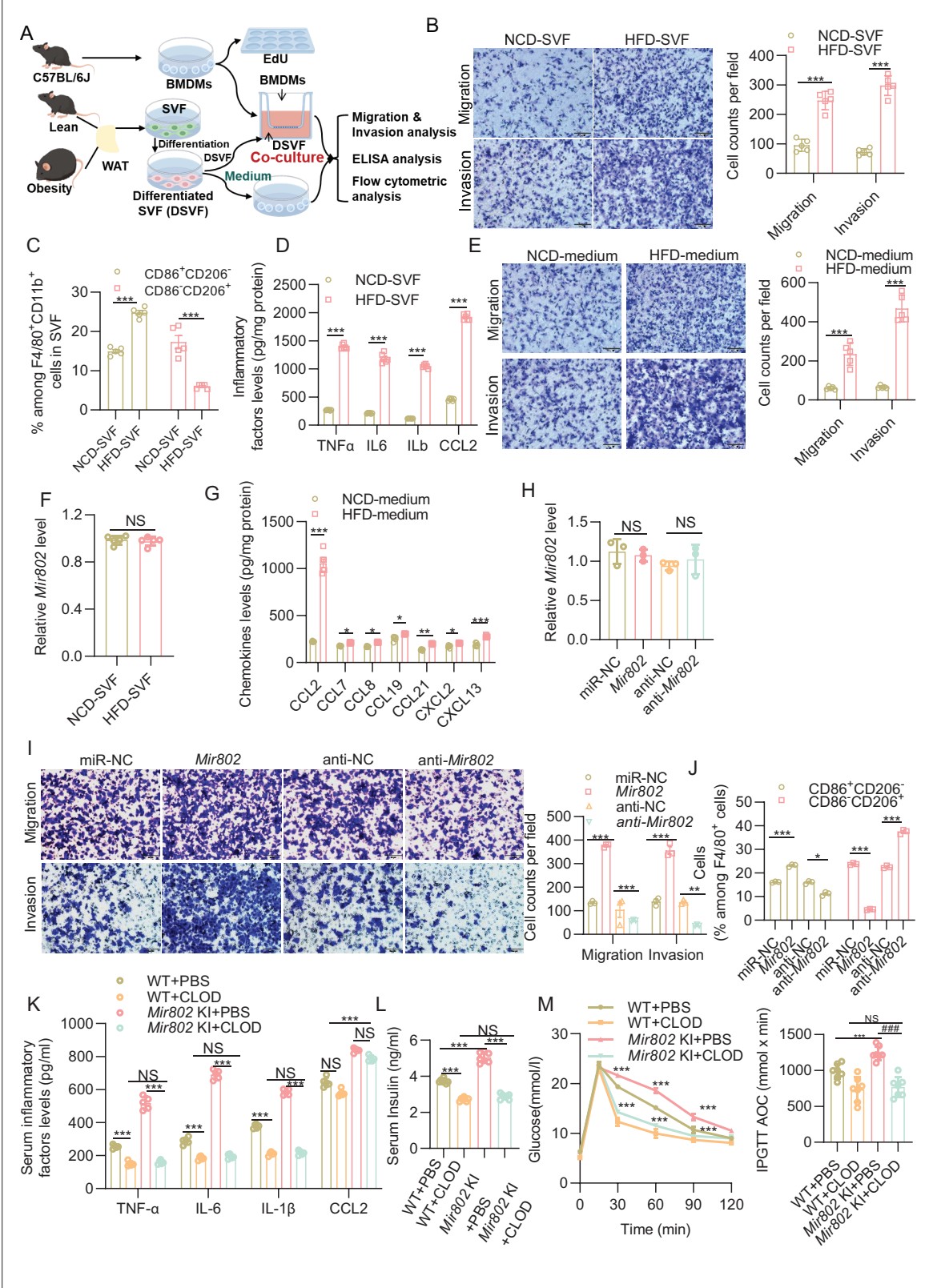

**Figure 4.** Interplay between Mir802 ectopically expressed adipocytes and macrophages. (**A**) Flowchart of the co-culture experiments designed for determining WAT SVF of obese adipose tissue can affect macrophages (bone marrow derived macrophages BMDMs). (**B**) Obesity promoted BMDMs migration and invasion in transwell migration and invasion assay. (**C**) M1 (CD86⁺CD206⁻) and M2 (CD206⁺CD86⁻) within the macrophage population. (**D**) The levels of TNF-α, IL-6, IL-1β, and CCL2 determined with ELISA. (**E**) Migration and invasion ability of BMDMs treated with a medium conditioned

*Figure 4 continued on next page*

*Figure 4 continued*

with obese or lean SVF cells. (**F**) The *Mir802* expression levels in the BMDMs after co-cultured WAT SVF cells. (**G**) Chemokine levels in the medium conditioned with obese or lean SVF cells. (**H**) The *Mir802* expression levels in the RAW264.7 cells after co-cultured with 3T3-L1 cells transfected with *Mir802* mimics or *Mir802* inhibitor. (**I**) *Mir802* induced 3T3-L1 cells recruitment more RAW 264.7 cells in transwell migration and invasion assay. (**J**) *Mir802* mimics-transfected 3T3-L1 cells promoted RAW 264.7 cells M1-like polarization. (**K–M**) Clodronate-conjugated liposomes (CLOD-liposomes) was injected into WT and *Mir802* KI mice. Serum levels of TNF-α, IL-6, IL-1β, CCL2 determined with ELISA n=5, (**K**); serum insulin levels tested with ELISA n=5, (**L**); glucose tolerance tested by IPGTT n=7, (**M**). Data represent mean ± SEM. Differences between groups were determined by ANOVA (**D–E, G–M**), or two-tailed unpaired Student's *t* test (**F**). **p<0.01, ***p<0.001.

The online version of this article includes the following figure supplement(s) for figure 4:

**Figure supplement 1.** Interplay between Mir802 ectopically expressed adipocytes and macrophages.

**Figure supplement 2.** Interplay between Mir802 ectopically expressed adipocytes and macrophages.

inflammatory factors (TNF-α, IL-6, IL-1β, and CCL2), and the differences in these circulating factors except CCL2 between *Mir802* KI mice and WT littermates became indistinguishable after macrophage depletion, suggesting that the macrophages are not the main source of CCL2 (*Figure 4K*). Furthermore, macrophage depletion also led to recover hyperinsulinemia (*Figure 4L*) and HOMA-IR (*Figure 4—figure supplement 2D*) in *Mir802* KI mice after the treatment. Similarly, Macrophage depletion with CLOD-liposomes obviously alleviated the HFD-induced glucose intolerance in *Mir802* KI mice, and abrogated the differences of these metabolic indicators between *Mir802* KI mice and WT mice (*Figure 4M*). Taken together, these data support macrophages as an important mediator for adipose *Mir802*–induced systemic inflammation (*Figure 4—figure supplement 2E*).

## *miRNA-802* promotes adipose tissue inflammation and insulin resistance by targeting TRAF3

To better understand the role of *Mir802* in regulating macrophage-mediated adipose tissue inflammation and insulin resistance, we next set out to identify the target genes of *Mir802* in adipocytes. For that, we utilized RNA-sequencing of samples derived from the epiWAT of *Mir802* KI mice and their WT littermates. A total of 191 differentially expressed genes were identified. The cutoff criteria for significant differentially expressed genes were log fold change >2 and adjusted p-value <0.05. We identified 29 upregulated genes and 57 downregulated genes (*Figure 5A* left, *Supplementary file 1b*). Then, we combined the multiMiR database (*Ru et al., 2014*) with prediction programs (Target-Scan Release 7.0 and miRPathDB) to predict possible targets of *Mir802*. Among 18 tested potential targets, TNF-receptor-associated factor 3 (*Traf3*) was identified as a genuine target of *Mir802*, which was among the genes that were significantly downregulated in *Mir802* KI versus WT epiWAT (*Figure 5A* right, *Figure 5—figure supplement 1A*). Indeed, we observed that TRAF3 was decreased in both mRNA and protein levels in obese humans and in the WAT of HFD, *Lep^{ob/ob}* and *Lepr^{db/db}* mice (*Figure 5B*, *Figure 5—figure supplement 1B*). The targeting potential between *Mir802* and *Traf3* was also observed in *Mir802* KI and *Mir802* KO mice (*Figure 5D*, *Figure 5—figure supplement 1D*). We then demonstrated *Mir802* binding to the *Traf3* 3'-UTR by transiently co-expressing luciferase reporter fusions of *Traf3* and *Mir802* mimics in 3T3-L1 cells. The results of these co-transfection experiments indicated that the relative luciferase activity in *Traf3* 3'-UTR-expressing cells was significantly inhibited by *Mir802*, whereas other *Traf3* 3'-UTR fusions that contained mutations (*Traf3*-MUT) in *Mir802* binding sites were unaffected (*Figure 5E*). Consistent with these findings, the ectopic expression of *Mir802* in 3T3-L1 cells effectively regulated the mRNA and protein levels of endogenous *Traf3* (*Figure 5F*, *Figure 5—figure supplement 1E*). Moreover, we conducted anti-Ago2 RIP in 3T3-L1 cells, which transiently overexpressed *Mir802*. Endogenous *Traf3* pulldown by Ago2 was specifically enriched in *Mir802*-transfected cells (*Figure 5G*) and vice versa (*Figure 5—figure supplement 1F*). Overall, these data suggest that *Traf3* is a direct target of *Mir802*.

To address whether the increase in inflammation and insulin resistance in *Mir802* KI mice was attributable to decreased *Traf3*, 8-week-old male *Mir802* KI mice were given AAV-*Adipoq-Traf3* (*Mir802* KI & *Traf3* eWAT OE) through epididymal fat pad. At 1 week after injection of adeno-associated virus (AAV) expressing *Traf3*, *Traf3* expression in the epiWAT of *Mir802*-KI mice was increased to a level similar to that in WT mice (*Figure 5H*). Notably, upregulation of *Traf3* led to significant decreases in the counts of total macrophages (*Figure 5J*, *Figure 5—figure supplement 1G*) and M1 macrophages (*Figure 5J*, *Figure 5—figure supplement 1H*) in the epiWAT of HFD-fed *Mir802* KI mice compared with

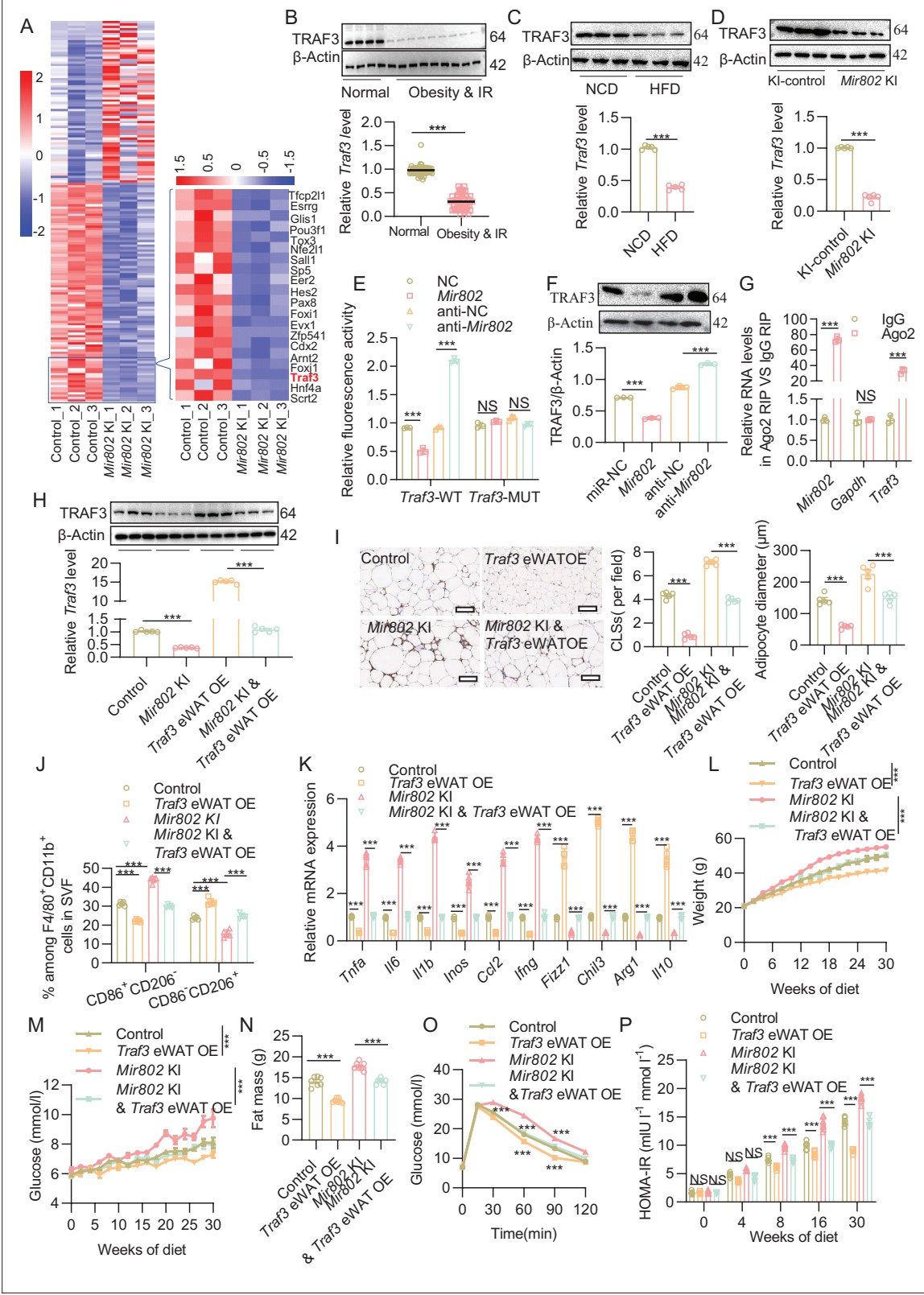

**Figure 5.** Adipose *Mir802* modulates recruitment and polarization of macrophages by directly targeting *Traf3*. (**A**) Heat map illustrating the differential expression of mRNAs in the epiWAT of *Mir802* KI mice compared to their WT *Mir802*^ki/ki littermates (n=3). (**B**) mRNA and protein levels of TRAF3 in human subcutaneous adipose tissues from obese and normal individuals ($n_{normal}$ = 4 and $n_{obesity\&IR}$=9). (**C, D**) mRNA and protein levels of TRAF3 in the epiWAT of HFD mice (C, n=3–5) or *Mir802* KI mice (D, n=3–5). (**E**) Relative luciferase activity in 3T3-L1 cells co-transfected with *Mir802* mimics and a

*Figure 5 continued on next page*

*Figure 5 continued*

luciferase reporter containing either *Traf3*-WT or *Traf3*-MUT. Data are presented as the relative ratio of Renilla luciferase activity to firefly luciferase activity. (**F**) Protein levels of TRAF3 in 3T3-L1 cells transfected with *Mir802* mimics or *Mir802* inhibitor. (**G**) Anti-Ago2 RIP was performed in 3T3-L1 cells transiently overexpressing *Mir802*, followed by qRT-PCR to detect *Traf3* associated with Ago2 (nonspecific IgG served as a negative control). (**H**) mRNA and protein levels of TRAF3 in the epiWAT of control, *Mir802* KI, *Traf3* eWAT OE, and *Mir802* KI and *Traf3* eWAT OE mice (n=3–5). (**I**) Representative images of F4/80 staining (left), quantification of CLSs (middle) and adipose diameter (right, n=5). (**J**) M1 (CD86⁺CD206⁻) and M2 (CD206⁺CD86⁻) within the macrophage population (n=5). (**K**) qRT-PCR analysis of the mRNA levels of M1 and M2 markers in the epiWAT of HFD-fed control, *Traf3* eWAT OE *Mir802* KI, and *Mir802* KI & *Traf3* eWAT OE(n=5). (**L, M**) Dynamic changes in body weight (**L**), glucose level (**M**), fat mass (**N**), glucose tolerance (**O**), and HOMA-IR (**P**) of control, *Mir802* KI, *Traf3* eWAT OE and *Mir802* KI and *Traf3* eWAT OE mice during 30 weeks of HFD feeding (n=7). Data represent mean ± SEM. Differences between groups were determined by ANOVA (**E–P**). ***p<0.001. *Mir802* abundance was normalized to *U6* level, and other genes levels were normalized to *Rn18s* abundance.

The online version of this article includes the following source data and figure supplement(s) for figure 5:

**Source data 1.** Related to *Figure 5B*.

**Source data 2.** Related to *Figure 5C*.

**Source data 3.** Related to *Figure 5D*.

**Source data 4.** Related to *Figure 5F*.

**Source data 5.** Related to *Figure 5H*.

**Figure supplement 1.** Adipose Mir802 modulates recruitment and polarization of macrophages by directly targeting Traf3.

**Figure supplement 1—source data 1.** Related to *Figure 5—figure supplement 1B*.

**Figure supplement 1—source data 2.** Related to *Figure 5—figure supplement 1C*.

**Figure supplement 1—source data 3.** Related to *Figure 5—figure supplement 1D*.

those treated with AAV8-vector. Coherently, the increased expression of M1 macrophage-associated proinflammatory factors (*Tnfa*, *Il6*, *Inos*, *Il1b*, and *Ifng*) in the epiWAT of HFD-fed *Mir802* KI mice was reversed by the AAV-mediated upregulation of *Traf3* (*Figure 5K*). In addition, *Traf3* eWAT OE reversed the weight gain (*Figure 5L*), hyperglycemia (*Figure 5M*), and adiposity (*Figure 5N*) induced by over-expression of *Mir802*. MRI analysis further confirmed that *Traf3* can reverse the increase in visceral fat caused by *Mir802* (*Figure 5—figure supplement 1I*). Consistent with these findings, upregulation of *Traf3* led to restoration of glucose intolerance (*Figure 5O*) and insulin resistance (*Figure 5—figure supplement 1J*) after 16 weeks of *Traf3* eWAT treatment in HFD-fed *Mir802* KI mice, coupled with a decrease in fasting insulin levels (*Figure 5—figure supplement 1K*) and ameliorative HOMA-IR (*Figure 5P*). Taken together, these findings support the notion that elevated *Mir802* induces macro-phage recruitment and polarization at least partly via downregulation of *Traf3*, thereby leading to adipose tissue inflammation and insulin resistance.

## *Mir802* activates noncanonical and canonical NF-κB pathways leading to macrophage recruitment

To further unravel the mechanism by which inhibition of TRAF3 expression induces adipose tissue inflammation, we looked for TRAF3 downstream cascades. Several studies have suggested that TRAF3 negatively regulates the noncanonical NF-κB pathway (*Liao et al., 2004*; *He et al., 2004*; *He et al., 2007*), which is consistent with the KEGG analysis based on our RNA-seq results (*Figure 6—figure supplement 1A*). This prompted us to measure NF-κB inducing kinase (NIK) protein levels and to explore the processing of p100 to p52. To test whether *Mir802* is required for the suppression of NIK protein levels, western blot analysis of NIK was performed on *Mir802*-overexpressed 3T3-L1 cells and *Mir802* ectopically expressed adipose tissue. As shown in *Figure 6A and B*, profound accumulation of NIK was observed in all cells with overexpression of *Mir802*, which correlated well with decreased TRAF3. *Mir802* selectively ablated adipose tissues showed the opposite result (*Figure 6—figure supplement 1B*). Processing of the p100 precursor to p52, the hallmark of noncanonical NF-κB activation, was also assessed by immunoblotting. Although 3T3-L1 cells exhibited the normal kinetics of p100 processing with substantial p52 accumulation by 48 hr after treatment with the empty vector, *Mir802*-overexpressing 3T3-L1 cells and *Mir802* selectively overexpressed adipose tissues showed constitutive and total processing of the p100 precursor protein (*Figure 6C and D*). On the contrary, there was less accumulation of p52 in *Mir802* selectively deleted adipose tissue (*Figure 6—figure supplement 1C*). As expected, IKK-α phosphorylation levels were also enhanced in 3T3-L1 cells and

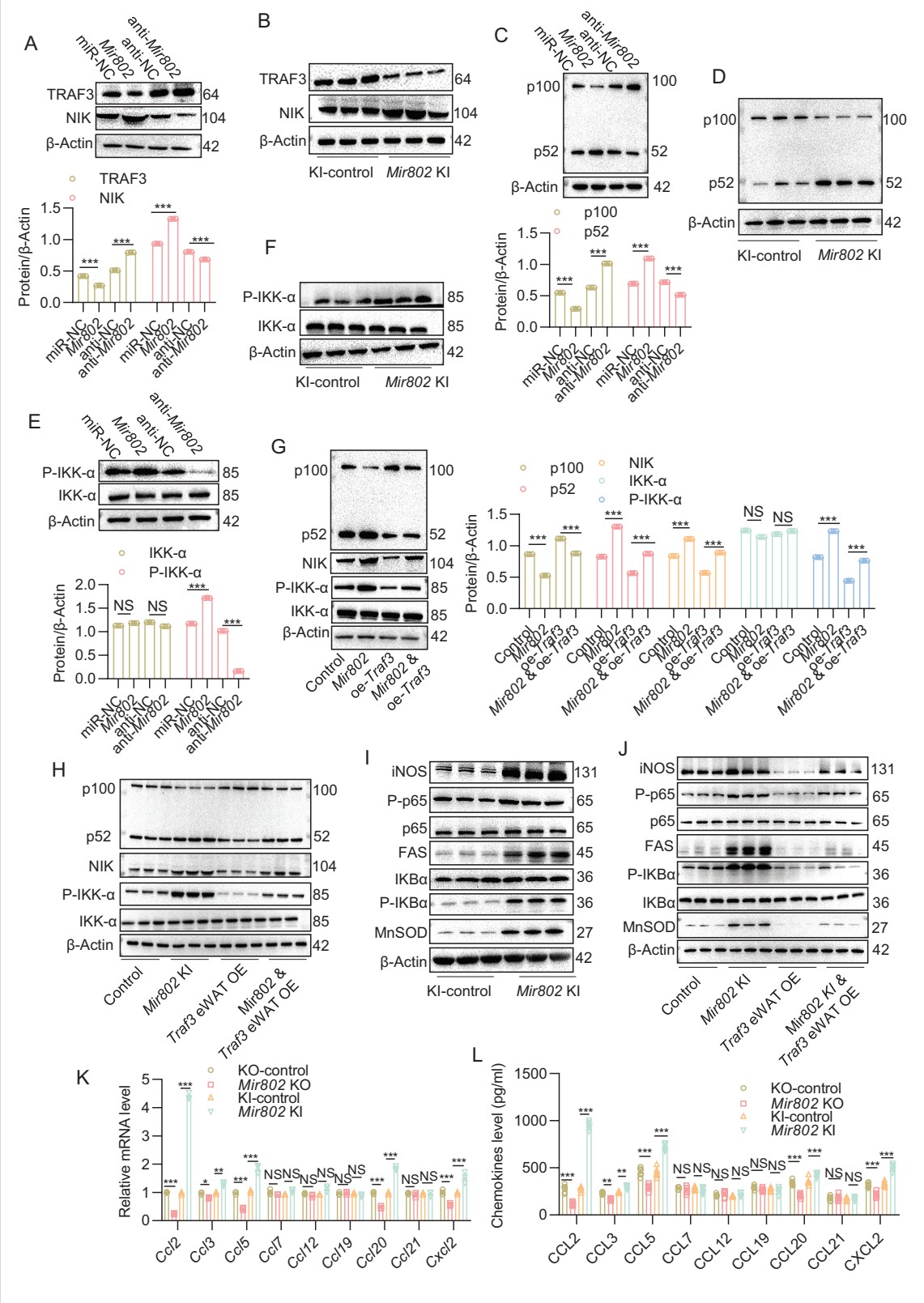

**Figure 6.** *Mir802* activates noncanonical and canonical NF-κB pathways by recruiting macrophages. (**A, B**) NIK protein levels in 3T3-L1 cells transfected with *Mir802* mimics or *Mir802* inhibitor (**A**) in the epiWAT of *Mir802* KI mice (B, n=3). (**C, D**) P100/52 protein levels in 3T3-L1 cells transfected with *Mir802* mimics or *Mir802* inhibitor (**C**), and in the epiWAT of *Mir802* KI mice (D, n=3). (**E, F**) Protein levels of IKK-α and P-IKK-α in 3T3-L1 cells transfected with *Mir802* mimics or *Mir802* inhibitor (**E**) in the epiWAT of *Mir802* KI mice (F, n=3). (**G, H**) Overexpression of *Traf3* reverses the protein levels of NIK,

*Figure 6 continued on next page*

*Figure 6 continued*

P-IKK-α, and P100/52 in 3T3-L1 cells (**G**) and in the epiWAT of *Mir802* KI mice (H, n=3). (**I, J**) Protein levels of some major canonical NF-*κ*B signaling targets in the epiWAT of *Mir802* KI mice (I, n=3) and *Traf3* eWAT OE rescued mice (J, n=3). (**K, L**) qRT-PCR (**K**) and ELISA (**L**) were performed to detect major chemokine levels. Data represent mean ± SEM. Differences between groups were determined by ANOVA (K–L). ***p<0.001. Genes levels were normalized to *Rn18s* abundance.

The online version of this article includes the following source data and figure supplement(s) for figure 6:

**Source data 1.** Related to *Figure 6A*.

**Source data 2.** Related to *Figure 6B*.

**Source data 3.** Related to *Figure 6C*.

**Source data 4.** Related to *Figure 6D*.

**Source data 5.** Related to *Figure 6E*.

**Source data 6.** Related to *Figure 6F*.

**Source data 7.** Related to *Figure 6G*.

**Source data 8.** Related to *Figure 6H*.

**Source data 9.** Related to *Figure 6I*.

**Source data 10.** Related to *Figure 6J*.

**Figure supplement 1.** Mir802 activates noncanonical and canonical NF-*κ*B pathways by recruiting macrophages.

**Figure supplement 1—source data 1.** Related to *Figure 6—figure supplement 1B*.

**Figure supplement 1—source data 2.** Related to *Figure 6—figure supplement 1C*.

**Figure supplement 1—source data 3.** Related to *Figure 6—figure supplement 1D*.

in the epiWAT of *Mir802* KI mice (*Figure 6E*, *Figure 6—figure supplement 1D*). To confirm that *Mir802* activates the noncanonical NF-κB pathway through TRAF3, *Traf3* plasmid was transfected into *Mir802*-overexpressing 3T3-L1 cells, then NIK protein levels and processing of p100 to p52 were again assessed by immunoblotting. As shown in *Figure 6G*, *Traf3* restored the levels of NIK and the processing of p100 to p52 in *Mir802*-overexpressing 3T3-L1 cells. Moreover, these results were confirmed in *Mir802* KI mice (*Figure 6H*), indicating that *Mir802* regulated the noncanonical NF-κB pathway via TRAF3.

Previous studies have suggested that TRAF3 also suppresses activation of the canonical NF-κB pathway (*Zarnegar et al., 2008*; *Bista et al., 2010*). To verify whether *Mir802* can regulate the canonical NF-κB pathway through TRAF3, nuclear extract was harvested from the adipose tissue of 16-week-old WT and mice with adipose-selective forced expression of *Mir802*. NF-κB activation status was then assessed by measuring p65, IκBα, and some major targets associated with the pathway. As shown in *Figure 6I* and p65 and IκBα were phosphorylated, and some major targets of canonical NF-κB signaling, such as MnSOD, FAS, and iNOS, were activated in *Mir802* KI mice. As expected, the activation of canonical NF-κB signaling in *Mir802* KI mice was partially reversed by overexpression of *Traf3* (*Figure 6J*). To assess the potential impact of heightened NF-κB activity, we harvested mRNA from the adipose tissue of WT and *Mir802* KI mice and analyzed the expression levels of multiple noncanonical and canonical NF-κB pathway target genes (*Akhter et al., 2021*; *Cildir et al., 2016*) using qRT-PCR. Here, we observed that the expression levels of CCL2, CCL3, CCL5, CCL20, and CXCL2 were elevated in the adipose tissue of *Mir802* KI mice (*Figure 6K*), which is consistent with ELISA results using the serum of *Mir802* KI mice (*Figure 6L*). Taken together, these data indicate that *Mir802* activates the noncanonical and canonical NF-κB pathways via TRAF3, leading to macrophage recruitment.

### *Mir802* promotes lipid synthesis and M1 macrophage polarization in adipose tissue through activating SREBP1

The mechanism of how *Mir802* increasing M1 polarization by inducing NF-κB pathways remains unclear. To better understand the role of *Mir802* in regulating macrophage polarization, we performed transcriptome sequencing using epiWAT derived from *Mir802* KI mice. In the adipose tissues of *Mir802* KI mice, the expression of 191 mRNAs was significantly altered compared to that of mRNAs in WT mice, of which the expression of 75 mRNAs increased (*Figure 7A*, *Supplementary file 1b*). We found

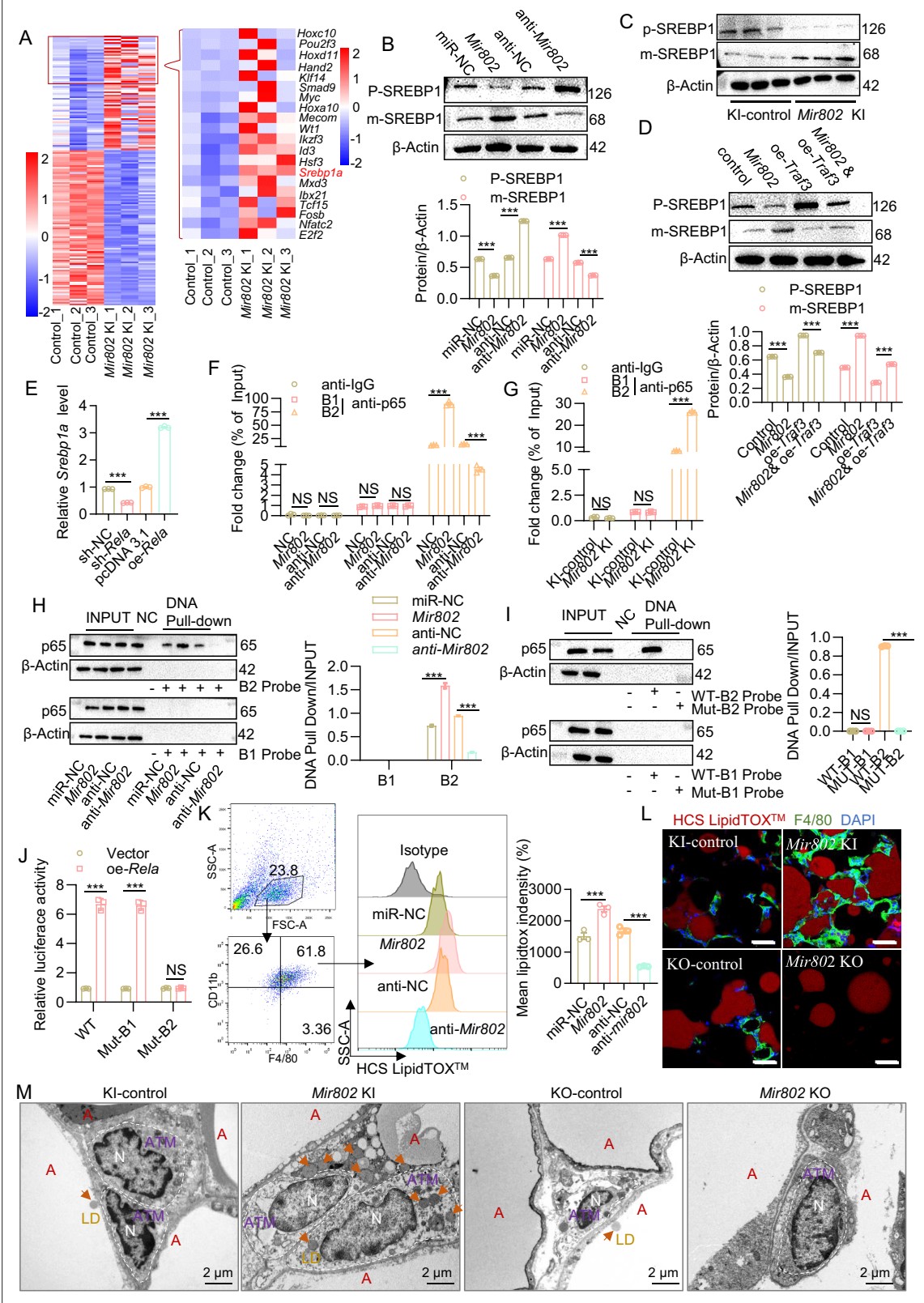

**Figure 7.** *Mir802* promotes lipogenesis and induces M1 macrophage polarization in adipose tissue through activating SREBP1. (**A**) Heat map illustrating the top 20 upregulated mRNAs in the epiWAT of *Mir802* KI mice compared to their WT *Mir802*fl/fl littermates (n=3). (**B–C**) Protein levels of the mature form of SREBP-1 protein (m-SREBP1) and the precursor form of SREBP-1 (P-SREBP1) in mature 3T3-L1 cells transfected with *Mir802* mimics or *Mir802* inhibitor (**B**), in the epiWAT of *Mir802* KI mice (**C**, n=3). (**D**) The protein levels of m-SREBP1 and P-SREBP1 were reversed by *Traf3*. (**E**) *Srebp1a* mRNA

*Figure 7 continued on next page*

*Figure 7 continued*

levels in 3T3-L1 cells transfected with *Rela*-overexpressing plasmid or *Rela* shRNA plasmid. (**F, G**) ChIP-qPCR assays were conducted to verify that *Rela* binds to the *Srebp1* promoter in 3T3-L1 cells transfected with *Mir802* mimics or *Mir802* inhibitor (**F**) and in the epiWAT of *Mir802* KI mice (**G**, n=3). (**H**) DNA pull-down assay using a biotinylated DNA probe corresponding to the −360 to −400 or −1198 to −1237 region of the *Srebp1* promoter in 3T3-L1 cells transfected with *Mir802* mimics or *Mir802* inhibitor. (**I**) DNA pull-down assay using a biotinylated DNA probe corresponding to the −1198 to −1237 region of the wild-type (WT) or a mutant sequence of the *Srebp1* promoter in 3T3-L1 cells stimulated with *Rela* plasmid for 48 hr. (**J**) Luciferase reporter assays in 3T3-L1 cells transfected with the indicated plasmids for 48 hr. Dual-luciferase activity was determined. (**K**) *Mir802* mimics or *Mir802* inhibitor was transfected into 3T3-L1 cells, then direct contact co-culture with mature 3T3-L1 and RAW264.7 cells, flow cytometry analysis of cellular neutral lipid content using the LipidTOX in cells. (**L**) Representative images of the immunofluorescence of lipid droplets (HCS LipidTOX, Red) and F4/80 (Green, n=3). Scale bar: 20 µm. (**M**) Transmission electron microscopy (TEM) was performed to detect the contact between lipid droplets and macrophages (n=3). Data represent mean ± SEM. Differences between groups were determined by ANOVA (E–G and J). ***p<0.001. Genes levels were normalized to *Rn18s* abundance.

The online version of this article includes the following source data and figure supplement(s) for figure 7:

**Source data 1.** Related to *Figure 7B*.

**Source data 2.** Related to *Figure 7C*.

**Source data 3.** Related to *Figure 7D*.

**Source data 4.** Related to *Figure 7H*.

**Source data 5.** Related to *Figure 7I*.

**Figure supplement 1.** Mir802 promotes lipogenesis and induces M1 macrophage polarization in adipose tissue through activating SREBP1.

**Figure supplement 1—source data 1.** Related to *Figure 7—figure supplement 1D*.

**Figure supplement 1—source data 2.** Related to *Figure 7—figure supplement 1E*.

that seven mRNAs were upregulated more than tenfold (*Figure 7A*, right), and these mRNAs were annotated using UCSC and Ensemble. Among these upregulated genes, we focused on the lipogenic gene sterol regulatory element-binding protein 1 a (SREBP1a), which is involved in fatty acid synthesis and lipid droplet formation (*Shimano and Sato, 2017*). We verified that the mRNA levels of *Srebp1a* were increased in both the epiWAT of *Mir802* KI mice and in 3T3-L1 cells transfected with *Mir802* mimics in qRT-PCR analysis (*Figure 7—figure supplement 1A* and B). We also found that the mature form of the SREBP-1 protein (m-SREBP1) was significantly higher in the epiWAT of *Mir802* KI and 3T3-L1 cells transfected with *Mir802* mimics (*Figure 7B and C*). As expected, overexpression of *Traf3* reduced the upregulation level of mature SREBP1 induced by *Mir802* (*Figure 7D*). However, target gene prediction algorithms as well as luciferase reporter and Ago2-RIP assays confirmed that *Srebp1a* is not the direct target gene for *Mir802* (data not shown). This prompted us to verify whether *Srebp1a* is a downstream gene in the NF-κB pathway.

For this purpose, we first predicted the nuclear factor-κB (NF-κB) family (*Rela*, *Relb*, *Rel*, *Nfκb1*, and *Nfκb2*) sites in the promoter of *Srebp1a* using JASPAR and the Promo database. Two *Rela* potential binding sites (B1 and B2) were found in the promoter of *Srebp1a* (*Figure 7—figure supplement 1C*). We next overexpressed *Rela* in 3T3-L1 cells; qRT-PCR results showed that *Rela* could increase *Srebp1a* expression in 3T3-L1 cells (*Figure 7E*). To determine whether *Srebp1a* is a direct *Rela* target gene, ChIP-qPCR assays were used. The results showed that occupancy of *Rela* binding site 2 (B2) on the *Srebp1a* promoter was significantly increased in *Mir802*-overexpressing 3T3-L1 cells and *Mir802* selectively overexpressed adipose tissues (*Figure 7F and G*, *Figure 7—figure supplement 1D*). We then conducted DNA pull-down assays to examine the binding of *Rela* to the *Srebp1a* promoter in vitro. We constructed two DNA probes containing −360 to −400 or −1198 to −1237 regions that contained the predicted binding site 1 (B1) and predicted binding site 2 (B2), respectively, to detect binding to *Rela* in nuclear extracts. Similar findings were obtained in that the B2 DNA probe, but not the B1 DNA probe, bound to *Rela* in the 3T3-L1 cell line overexpressing *Mir802* (*Figure 7H*). However, with mutant B2 (agggaatgct, Mut2), DNA pull-down results showed that *Rela* could not bind to Mut2 (*Figure 7I*). Moreover, we constructed a luciferase reporter plasmid containing the *Srebp1a* promoter region from −1295 to +1 WT and two mutant reporter plasmids mutated in −375 to −385 (Mut1) or in −1211 to −1221 (Mut2). Overexpression of *Rela* significantly enhanced WT and Mut1, but not Mut2-driven activity luciferase in 3T3-L1 cells (*Figure 7J*). Taken together, these results indicated that *Mir802* indirectly stimulates *Srebp1a* expression via the canonical NF-κB signaling pathway (*Figure 7—figure supplement 1F*).

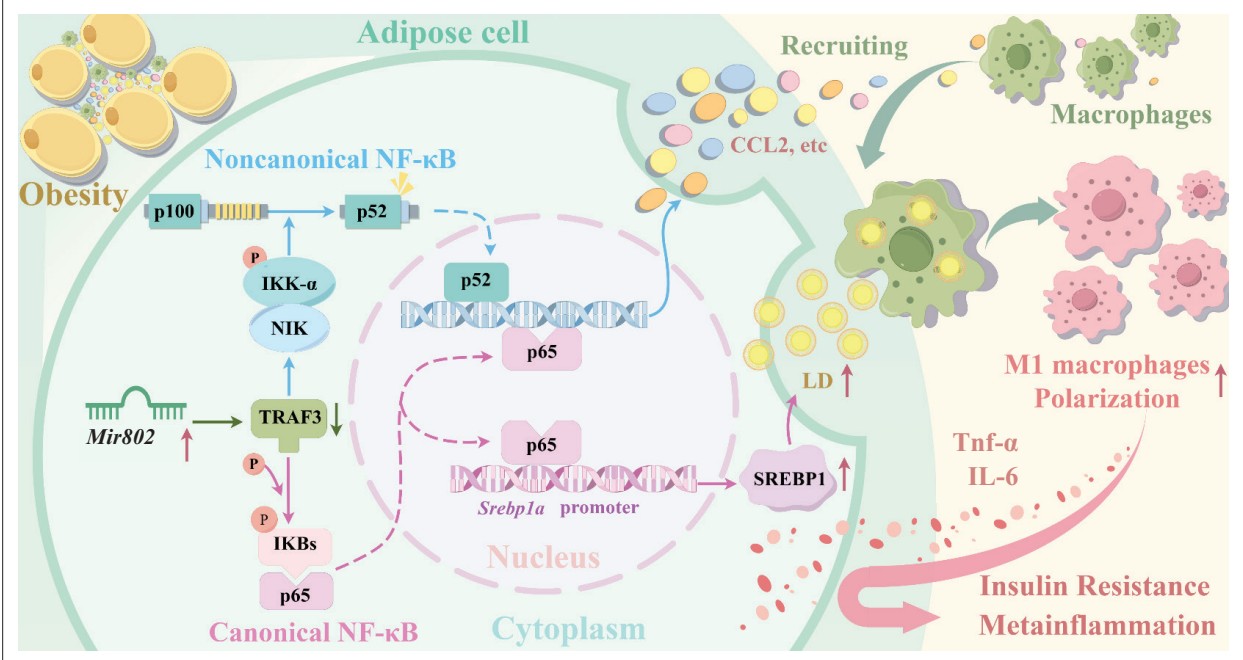

**Figure 8.** Schematic illustration for the mechanism of *Mir802* exacerbates adipose tissue inflammation and leads to metabolic dysfunction during obesity. We found that *Mir802* endows adipose tissue with the ability to interact with macrophages and regulate the inflammatory cascade. During obesity, *Mir802* promotes adipose tissue secretion more chemokines to recruit macrophages by targeting *Traf3 activating* canonical and noncanonical NF-$\kappa$B signaling pathways; and *Mir802* increases lipogenesis through promoting *Srebp1* transcription, leading macrophages toward proinflammatory M1 phenotype by engulfing lipid droplets. Schematic illustration was drawn by figdraw.

As previously described, *Srebp1a* is a well-established regulator of lipid synthesis (**Shimano and Sato, 2017**). Accordingly, *Mir802* overexpression significantly increased the number and fluorescence intensity of lipid droplets, which correlated well with increased SREBP1. Conversely, knockdown of *Mir802* strongly reduced lipid droplet formation in 3T3-L1 cells and *Mir802*-KO mouse adipose tissue (**Figure 7—figure supplement 1G** and H). Lipid droplets have been shown to play a crucial role in M1 macrophage polarization (**Lumeng et al., 2007a**; **Prieur et al., 2011**). Consistent with this, we found overexpression-*Mir802* 3T3-L1 cells promoted RAW264.7 cells to engulf more lipid droplets through direct contact co-culture with mature 3T3-L1 and RAW264.7 cells, while the content of lipid droplets is decreased in the knockdown-*Mir802* 3T3-L1 cells (**Figure 7K**). Moreover, we also observed that adipose tissue macrophages (ATMs) of the *Mir802* KI mice could engulf more lipid droplets (**Figure 7L and M**). The elevated expression of the classical activation marker further indicated that lipid droplets induced the ATMs in *Mir802* KI mice to the pro-inflammatory phenotype (**Figure 7—figure supplement 1I**). Altogether, these data show that *Mir802* indirectly regulates lipid droplet formation through SREBP1 and ultimately promotes macrophage M1 polarization.

## Discussion

Macrophage infiltration of adipose tissue has been described in both mice and humans during obesity. However, how lipid-loaded hypertrophic adipocytes send signals to trigger infiltration and alter the polarization of macrophages in obesity remains poorly understood. In this study, we found that *Mir802* endows adipose tissue with the ability to interact with macrophages and regulate the inflammatory cascade. Mechanistically, *Mir802* promotes adipose cells to secrete CCL2 recruiting macrophages, and *Mir802* increases lipid synthesis and promote macrophage to engulf lipids driving the polarization program toward proinflammatory M1 phenotype by targeting the cytoplasmic adaptor protein TRAF3 (**Figure 8**). Our findings indicate that *Mir802* has essential roles in the initiation and maintenance of adipose tissue inflammation and systemic insulin resistance.

Adipose tissue inflammation is a hallmark of obesity and a causal factor of metabolic disorders such as insulin resistance. Mice fed an HFD frequently develop chronic low-grade inflammation within

adipose tissues, characterized by increased infiltration of macrophages and the production of pro-inflammatory cytokines. Here, we showed that the increasing trend of *Mir802* in adipocytes is an early event during the development of adipose tissue obesity induced by an HFD. *Mir802* expression in visceral fat was progressively increased with the development of dietary obesity, whereas adipose-selective ablation of *Mir802* protected mice from exacerbation of meta-inflammation and insulin resistance caused by dietary stress. The high level of *Mir802* expression in visceral fat may partly explain why this adipose depot is more prone to inflammation and is closely related to insulin resistance. *Mir802* is required for adipose tissue inflammation and has major roles in macrophage recruitment and polarization. Thus, *Mir802* is crucially involved in initiating inflammatory cascades in obese adipose tissue. Moreover, the finding that *Mir802* inhibitor treatment ameliorated pre-established adipose inflammation in DIO mice indicates that *Mir802* is also essential for maintenance of the inflammatory response. Although previous studies have found that *Mir802* was up-regulated in the adipose tissue during obesity (*Zhang et al., 2020*; *Kornfeld et al., 2013*; *Ge et al., 2022*; *Goga et al., 2021*), the function of *Mir802* was focused on cancers (*Gao et al., 2021*), liver (*Seok et al., 2021*; *Ni et al., 2021*), small intestine (*Goga et al., 2021*), and pancreas (*Ge et al., 2022*). And *Yao et al., 2020* have determined that up-regulation of *Mir802* exacerbates inflammatory process of inflammatory bowel disease via targeting *Socs5*, which indicates that *Mir802* play an important role in regulating the inflammatory process. Whether *Mir802* can regulate adipose function is still confused, our study is the first to address that *Mir802* intensifies adipose tissue inflammation by affecting crosstalk between adipose and macrophage. The findings that systemic insulin resistance is ameliorated by *Mir802* depletion and is aggravated by adoptive transfer of *Mir802* mimics strongly suggest that *Mir802*-dependent adipose inflammation has an impact on systemic metabolism.

Like most other miRNAs, *Mir802* regulates the expression of multiple genes in different tissues. In the liver, *Mir802* is induced by obesity and impaired glucose tolerance, and it attenuates insulin sensitivity by downregulation of *Hnf1b* (*Kornfeld et al., 2013*). Genetic ablation of *Mir802* in the small intestine of mice leads to decreased glucose uptake, impaired enterocyte differentiation, increased Paneth cell function, and intestinal epithelial proliferation through derepression of *Tmed9* (*Goga et al., 2021*). We recently discovered that in pancreatic islet cells, elevated *Mir802* causes impaired insulin transcription and secretion by targeting *Neurod1* and *Fzd5* (*Zhang et al., 2020*). In this study, we found that *Mir802* promotes adipose tissue inflammation and insulin resistance by targeting TRAF3 in adipocytes. As a member of the TNF receptor (TNFR) superfamily, TRAF3 plays vital roles in inflammatory responses via activation of both the canonical and noncanonical NF-κB signaling pathways *He et al., 2007*; *Zarnegar et al., 2008* following engagement of a variety of TNFR superfamily members such as Baff receptor, lymphotoxin β receptor, and CD40 (*Häcker et al., 2011*). Here, we found that *Mir802* can regulate the NF-κB pathway by directly targeting TRAF3 rather than by activating the classic receptor, which enriches the understanding of the NF-κB pathway.

Macrophage accumulation was significantly higher in adipose tissue from HFD-fed *Mir802* KI mice than in WT mice, suggesting that overexpression of *Mir802* enhances the infiltration ability of macrophages. Correspondingly, we observed that *Mir802*-overexpressing adipocytes released more chemokines by activating NF-κB pathway, such as CCL2, CCL5, CCL20, and CXCL2. Adipose tissue inflammation is well documented as an important contributor to systemic insulin resistance (*Lumeng et al., 2007b*). This was further validated by our observation of enhanced adipose tissue inflammatory responses in *Mir802* KI mice. Moreover, HFD-fed *Mir802* KI mice exhibited adipose tissue macrophage infiltration, proinflammatory cytokine expression, and NF-κB pathway activation. Genes that are crucial for meta-inflammation and insulin resistance were directly affected by the enhancement of *Mir802* in adipose tissue. Thus, increased adipose tissue inflammation resulting from *Mir802* overexpression contributed, in large part, to systemic insulin resistance in *Mir802* KI mice.

The chronic inflammation microenvironment is one of the main features of obesity. A recent study found that adipocytes can release lipid-filled vesicles that become a source of lipids for local macrophages (*Flaherty et al., 2019*). Phagocytosis or excessive accumulation of lipid droplets can induce macrophage M1 polarization (*Flaherty et al., 2019*; *Batista-Gonzalez et al., 2019*). In our study, we observed the same phenomenon, that is, in *Mir802* KI mice, macrophages accumulated more lipid droplets and exhibited an inflammatory phenotype. SREBP1 has been found to promote the acute inflammatory response and lipogenesis (*Im et al., 2011*; *Fei et al., 2023*). Here, we found that *Mir802* increased SREBP1 expression inducing lipogenesis by activating canonical NF-κB signaling pathways,

then macrophage engulf lipid droplet promoting macrophage M1 polarization. This has enriched our understanding of the functionality of SREBP1 to some extent. However, *Mir802* only indirectly regulates SREBP1, but it still has a considerable impact on macrophages, indicating the importance of miRNA positive or indirect regulation. Though Jae-Ho Lee (*Lee et al., 2018*) determined that Srebp1a provides lipids essential for maintaining association between the actin cytoskeletal network and plasma membranes, thus enhancing phagocytosis, whether *Mir802* promotes macrophage to engulf lipid droplets via regulation the actin cytoskeletal network and plasma membranes still need to further research.

Taken together, our results support the idea that obese adipose tissue activates *Mir802*, which, in turn, initiates and propagates inflammatory cascades, including the recruitment of macrophages into obese adipose tissues and their subsequent induction of the inflammatory phenotype. Thus, *Mir802* appears to have a primary role in obese adipose tissue inflammation. In our study, we have found that a high-fat diet (nutrition factor) and genetic factors can induce an increase in adipose *Mir802*. Some environmental factors including circadian rhythms (*Chaix et al., 2019*) and exposure to Di (2-ethylhexyl) phthalate (DEHP; *Zhang et al., 2023*) are also important for obesity development. The current research lacks evidence on the impact of environmental factors on *Mir802* expression in adipose tissue. So future studies are needed to clarify which environmental cues such as within obese adipose tissue initiate *Mir802* elevation. The present observations indicate that *Mir802* inhibitors might offer a novel approach to prevent diseases associated with insulin resistance.

## Materials and methods
### Animal studies
All mice used were of mixed strain backgrounds with approximately equal contributions from C57BL/6 J, with the exceptions of $Lepr^{db/db}$ mice (C57BLKS/J). *Mir802* conditional knockout ($Mir802^{fl/fl}$) and knock-in ($Mir802^{ki/ki}$) mice were generated previously via CRISPR/Cas9 system from Model Animal Research Center of Nanjing University (Nanjing, China). The detailed information was initially described in *Zhang et al., 2020*. *Adipoq*-Cre BAC transgenic mice on a C57BL/6 J background were obtained from Jackson Laboratories (#028020). *Adipoq*-Cre intercrossed with $Mir802^{fl/fl}$ or $Mir802^{ki/ki}$ mice to generate $Mir802^{fl/+}$*:Adipoq*-Cre or $Mir802^{ki/+}$*:Adipoq*-Cre mice. $Mir802^{fl/+}$*:Adipoq*-Cre or $Mir802^{ki/+}$*:Adipoq*-Cre mice were then bred with $Mir802^{fl/fl}$ or $Mir802^{ki/ki}$ to generate $Mir802^{fl/fl}$*:Adipoq*-Cre/ $Mir802^{ki/ki}$*:Adipoq*-Cre (*Mir802* KO/*Mir802* KI) along with $Mir802^{fl/fl}$/ $Mir802^{ki/ki}$ and *Adipoq*-Cre mice. Genotyping was performed as described (*Zhang et al., 2020*) and the PCR primers were listed in *Supplementary file 1c*. Studies were performed on 8-week-old male and female mice initially housed under standard conditions with full access to standard mouse chow and water. After this time, mice were switched to a 60% high-fat diet (HFD, Research Diets, Cat# D12492) or normal chow diet (NCD, Research Diets, Cat#D12450J) consisting of a 10% fat diet for 30 weeks. All mice had free access to food and water ad libitum. Animals were housed in a temperature-controlled environment with a12 h dark–light cycle. At the end of the 30-week period, mice were euthanized via overdose of isoflurane anesthesia, and tissues were immediately weighed, dissected, and frozen in liquid nitrogen. Tissue samples were stored at –80 °C until use. Care of all animals was within institutional animal-care committee guidelines, and all procedures were approved by the animal ethics committee of China Pharmaceutical University (Permit Number: 2162326) and were in accordance with the international laws and policies (EEC Council Directive 86/609,1987).

For administration of AAV8-Adipoq-*Mir802* sponge vector, AAV8-Adipoq-*Traf3* vector to epididymal adipose tissue, mice were anesthetized with pentobarbital sodium (60 mg/kg) intraperitoneally and the laparotomy was performed. Each epididymal fat pad was given eight injections of 5 µl ($1\times10^{10}$ viral genome copies) of AAV solution.

### Human adipose samples of lean and overweight individuals
Adipose and clinicopathological data were collected from Sir Run Run Hospital, Nanjing Medical University (Nanjing, China). All patients enrolled in this study were obese (BMI >25 kg/m$^2$). The negative controls were normal-weight individuals (20≤BMI ≤ 25 kg/m$^2$). All human subjects provided informed consent. All human studies were conducted according to the principles of the Declaration

of Helsinki and were approved by the Ethics Committees of the Department Sir Run Run Hospital (Nanjing, China, 2023-SR-046). The clinical features of patients are listed in *Supplementary file 1a*.

## Adipose sample preparation

SVF and mature adipocytes were obtained as follows: adipose tissue samples were digested with collagenase type 1 in Krebs-RingerHenseleit (KRH) buffer for 30 min at 37 °C. Cell suspensions containing mature adipocytes and SVF were then filtered with nylon mesh and washed three times with KRH buffer. Mature adipose was floated to the surface and the remaining solution containing the SVF was centrifuged at 1500 rpm for 5 min. The pellet was washed with pre-adipocyte growth medium (DMEM-F12 supplemented with 10% calf serum and 1% penicillin-streptomycin), followed by a second centrifugation. SVF cells were then cryopreserved using a freezing medium (DMEM-F12 supplemented with 60% FBS and 10% DMSO). The medium was added to the pellet and frozen with a temperature gradient (−1 °C/min) and stored in liquid nitrogen until analysis. Following collection, whole adipose tissue samples were quickly frozen in liquid nitrogen and stored until analysis.

## 3T3-L1 cell culture and differentiation

3T3-L1 cells ($1 \times 10^5$ well$^{-1}$ in six-well plate) counted by blood counting chamber were cultured in DMEM (Gibco) containing 10% calf serum with high glucose at 37 °C, 5% $CO_2$ and full saturation humidity until they reached 80–90% confluence, at which point the media was changed to the first differentiation medium containing high glucose DMEM, 10% FBS, 0.5 mM 3-isobutyl-1-methylxanthine (IBMX), 1 µM dexamethasone and 10 µg/ml insulin for 48 hr, then the media was changed to the terminal differentiation cocktail containing high glucose DMEM, 10% FBS, and 10 µg/ml insulin for 48 hr obtained mature 3T3-L1 cells. 3T3-L1 cell line were purchased from the Chinese Academy of Sciences (Shanghai, China). Authentication of all cells was done by short tandem repeat (STR) profiling, confirming the absence of mycoplasma contamination.

## Insulin-resistant cell model

The insulin-resistant (IR) cell models were established in mature 3T3-L1 and mature WAT SVF cells by 0.5 mM palmitate acid (Cayman Chemical Company, Ann Arbor, MI, USA), 10 µg/ml insulin (Sigma-Aldrich) and 25 mM glucose for 24 hr. Relevant cellular samples were collected for subsequent qRT-PCR analysis.

## Bone-marrow-derived macrophages

For isolation of bone-marrow-derived macrophages (BMDMs), tibias and femurs were removed from 8-week-old mice and flushed with media using a 26-gauge needle. Bone marrow was collected at 500 × *g* for 2 min at 4 °C, resuspended with complete DMEM medium and filtered through a 70 µm cell strainer. We then obtained the nonadherent cells and suspended them in BMDM culture medium (DMEM supplemented with 10% fetal bovine serum, 20 ng/ml M-CSF) at a concentration of $10^7$ cells/ml in a total of 10 ml of medium. Fluid was changed every 2 days after inoculation until maturation on the 7th day.

## Migration and invasion assays

The 3T3-L1 cells or differentiated SVF cells were evenly plated in 24-well plates. To differentiate mature cells, migration and invasion assays were performed using a transwell chamber (Millipore, Billerica, MA, USA). For the migration assay, RAW 264.7 macrophage cells were seeded in the upper chamber with serum-free medium ($1.0 \times 10^5$ cells); the bottom chamber contained mature 3T3-L1 cells. For the invasion assay, the chamber was coated with Matrigel (BD Biosciences, Franklin Lakes, NJ, USA); the subsequent steps were similar to the migration assay. After the cells migrated or invaded for 24 hr, they were fixed and stained with crystal violet. Migrated and invaded RAW 264.7 cells were counted under an inverted light microscope. The number of migrated or invaded cells was quantified by counting the number of cells from 10 random fields at ×100 magnification.

## RNA-sequencing analysis

Total RNA from epididymis white adipose tissue of wild type control mice (n=3) and *Mir802* KI mice (n=3) was isolated using the RNeasy mini kit (QIAGEN) following the protocol. The quality of the

samples, the experiment, and the analysis data was completely finished by the HaploX (Shangrao, China). Cuffdiff (v2.2.1) 51 was used to calculate the fragments per kilobase million (FPKM) for mRNAs in each group. A difference in gene expression with a <i>P-value ≤ 0.05 was considered significant. The raw data is presented in *Supplementary file 1b*. The RNA-seq raw data that support the findings of this study has been deposited in the NCBI's Sequence Read Archive (SRA) database (PRJNA1021754).

## Fluorescence in situ hybridization (FISH)

Cy3 labeled *Mir802* probe was designed and synthesized by GenePharma (Shanghai, China). The frozen sections of adipose tissue from obese patients, normal persons or obese mice were fixed with 4% formaldehyde at room temperature for 10 min. The probe was hybridized at 37°C for 16 hr. DAPI was added at 1:5000 for 15 min after washing with probe detergent. Images were obtained with confocal laser scanning microscope (CLSM, LSM800, Zeiss, Germany) and processed using the ZEN imaging software.

## Plasmid and shRNA construction

The coding sequences for *Traf3* (NM_001286122.1), *Rel* (NM_ 009045.5), *Relb* (001290457.2), *Srebp1* (001313979.1) were amplified by PCR from full-length cDNA of mice, and then cloned in pcDNA 3.1 (+) vector (Addgene, Watertown, MA, USA). All plasmids were confirmed to be correct by sequencing. The primer sequences for PCR are listed in *Supplementary file 1c*.

The shRNA of *Traf3*, *Srebp1*, *Rela*, *Relb* were constructed in plvx-shRNA2 lentivirus vector (Takara). The plvx-shRNA2 lentivirus vector was digested with *EcoR I* and *BamH I*. The shRNA primer sequences are listed in *Supplementary file 1d*.

## Luciferase assay

*Mir802* mimics/*Mir802* inhibitor (anti-*Mir802*)/miRNA NC (NC)/miRNA inhibitor NC was purchased from GenePharma (Shanghai, China). The construction of *Traf3* (both wild type and mutants) was achieved by digestion of pmir-PGLO vector (Addgene, Watertown, MA, USA) with double restriction enzymes (*Xhol I* and *Xbal I*), followed by ligation of sequences encoding the corresponding 3'UTR of the target genes. Sequences of the synthetic oligonucleotides encoding the 3'UTR of the target genes and their mutants are listed in *Supplementary file 1c*. 3T3-L1 cells were transfected with one of the above-mentioned plasmids using Lipofectamine 2000 (Invitrogen), according to the manufacturer's instructions. At 48 hr after transfection, the cells were lysed and the luciferase activity was assayed with a dual-luciferase reporter assay kit (Vazyme, Nanjing, China). Data are presented as the ratio of Renilla luciferase activity to firefly luciferase activity.

## Flow cytometric analysis of macrophage polarization

SVF cells were stained with 50 µL Zombie NIR (1:200 in PBS for analysis on a Cytek Aurora spectral flowwas cytometer, #423105, Biolegend) at room temperature for 10 min. Afterward, the cells were washed once with FACS buffer (1% BSA in 1×PBS) and centrifuged at 500×*g* for 5 min at 4 °C. The cells were incubated with 50 µL of FcBlock (1:100, BD Pharmigen in FACS buffer) at room for 10 min, and then an equal volume of a 2×stain cocktail was added and mixed. For flow cytometry analysis of macrophages, 1×10⁶ freshly isolated cells were triple stained with CD11b-APC (#101211, Biolegend; 1:100), and F4/80-PE (#157304, Biolegend; 1:100), or stained with F4/80-FITC (#123108, Biolegend; 1:100), CD206-APC (#141708, Biolegend;1:100), CD86-PE (#105007, Biolegend; 1:100), or their isotype controls (Biolegend) on ice for 30 min in the dark. Samples were acquired on a 12-color Cytoflex (BD Biosciences) for each analysis. Dead cells were excluded from the analysis using Near-IR Live/Dead fixable staining reagent (BioLegend). Macrophages were identified as CD11b⁺F4/80⁺ and further as CD86⁺CD206⁻ or CD86⁻CD206⁺ cells within this gate. Data were analyzed using FlowJo software version X.0.7 (Tree Star, Inc).

## Mouse metabolic studies

After 12 hr fasting treatment, mice fasting blood glucose (FBG) levels were examined via using a glucometer (OMRON, Japan) and fasting serum insulin (FINS) levels were tested by insulin ELISA kit (Crystal Chem, USA). And the homeostatic model assessment indices of insulin resistance (HOMA-IR) was calculated with the equation (FBG (mmol/l)×FINS (mIU/l))/22.5.

During Glucose Tolerance Test (GTT), *Mir802* KO, *Mir802* KI and control group mice were fasted for 12 hr (overnight) and then intraperitoneally (i.p.) injected with 2 g/kg glucose (Sigma-Aldrich, StLouis, MO, USA) of body weight. Blood glucose was determined from tail vein blood at the indicated time (0, 15, 30, 60, 90, and 120 min) after glucose injection or oral using a glucometer (ACCU-CHEK Perfor, Roche).

For Insulin Tolerance Test (ITT), the mice were fasted for 6 hr before 0.75 U/kg human insulin (Roche) of body weight i.p. injected. Blood glucose was determined from tail vein blood at the indicated time (0, 15, 30, 60, 90, and 120 min) after insulin injection or oral using a glucometer (ACCU-CHEK Perfor, Roche). Subtracting the baseline area, by subtracting the starting glucose value from the value at each time point, generates the area of the curve (AOC; *Virtue and Vidal-Puig, 2021*).

## Body composition

The changes in body composition were assessed as we have previously described (*Gordon et al., 2020*). In brief, mice were anesthetized with 2% isoflurane by volume in a box and fixed on an MRI platform (Bruker BioSpec 7T/20 USR). Anesthesia was also maintained with isoflurane of 1% by volume. After turning on the instrument, the mice were scanned layer by layer according to the cross-section of their internal adipose tissue content, using ImageJ software analysis and statistics of the lipid distribution in mice.

## RNA isolation and qRT-PCR analysis

Total RNA from adipose tissues or its fractions or 3T3-L1 cells was extracted with TRIzol reagent (Invitrogen). For mRNA expression analysis, 500 ng of total RNA was used for synthesis of cDNA using PrimeScriptTM RT reagent Kit (Takara, Tokyo, Japan). For miRNA expression analysis, 150 ng total RNA was reverse-transcription into cDNA using miRNA-specific primers supplied with TaqMan MicroRNA Reverse Transcription kit. The quantitative real-time PCR was performed using the Light-Cycle 480 (Roche). The relative level of gene expressions were calculated by the $2^{-\Delta\Delta CT}$ method, after normalization with the abundance of *Rn18s* or *U6*. For *Mir802-5p* and *U6*, TaqMan probes (Ambion) were used to confirm our results. The sequences of genes were listed in *Supplementary file 1e*.

## Western blot analysis

Proteins were extracted from tissues or cells in radioimmunoprecipitation assay (RIPA) buffer (Beyotime) containing a complete protease inhibitor cocktail (Roche), resolved by SDSPAGE, transferred onto polyvinlidene fluoride (PVDF) membranes (Bio-Rad), and then probed with primary antibodies against TRAF3 (#ab36988), NIK (#ab314146) were from Abcam, NF-κB2 p100/p52 (#4882), NF-κB Rel (#8242) were from CST, and Relb (#A23389), NF-κB1 (#ab125611), IKK-α (#A2062), phospho-IKK-α (#AP0546), IKBα (#A19714), phosho-IKBα (#AP0614), iNOS (#A14031), phosho-Rel (#AP0123), Tubulin (#AC008), β-Actin (#AC038), and Histone H3 (#A2348) were from ABclonal. The protein bands were visualized with enhanced chemiluminescence reagents (GE Healthcare) and quantified by using the ImageJ software.

## RNA immunoprecipitation (RIP)

RNA immunoprecipitation was performed using an EZMagna RIP Kit (Millipore, Billerica, MA, USA) following the manufacturer's protocol. 3T3-L1 cells transfected with *Mir802* or oe-*Traf3* were lysed in complete RIP lysis buffer, and then, 100 µl of whole cell extract was incubated with RIP buffer containing magnetic beads conjugated with anti-Ago2 (#ab186733, Abcam) antibody or negative control normal mouse IgG (#ab172730, Abcam). Furthermore, purified RNA was subjected to qRT-PCR analysis to demonstrate the presence of the binding targets using the respective primers. The primer sequences are listed in *Supplementary file 1e*.

## Chromatin immunoprecipitation assay (ChIP)

ChIP experiments were strictly performed according to the manual for the ChIP Assay Kit (#17–10086, Millipore) and the manufacturer's protocol. 3T3-L1 cells transfected with *Mir802* mimics, *Mir802* inhibitor, oe-*Traf3*, or *Mir802* & oe-*Traf3* were fixed with 37% formaldehyde for 10 min, followed by 30 rounds of sonication, each for 3 s, to fragment the chromatin. The chromatin was incubated with NF-κB Rel antibody (#8242, CST) at 4 °C overnight and then immunoprecipitated with Proteinase K

(Millipore). Purified DNA was amplified by PCR using primer pairs that spanned the predicted *Rela* binding sites on the *Srebp1a* promoter. The primer sequences are listed in *Supplementary file 1e*.

### Agarose-oligonucleotide pull-down assay

The oligonucleotides for the mouse *Srebp1a* promoter and their complementary strands were synthesized by GenePharma (Shanghai, China) and biotinylated using a Pierce Biotin 3' End DNA Labeling Kit (Cat. #89818; Thermo Fisher Scientific). These oligonucleotides were annealed to form double-stranded oligonucleotides, which were then incubated with streptavidin-conjugated agarose beads at 4 °C for 60 min and washed twice with IP lysis buffer. Next, the nuclear extract (50 µg each) in 200 µl IP lysis buffer was pre-cleared with agarose beads at 4 °C for 90 min to reduce any nonspecific binding and then incubated with oligo/streptavidin-conjugated beads at 4 °C overnight. The mixtures were washed three times with IP lysis buffer via centrifugation the following day, and the affinity-purified proteins were eluted by boiling in SDS sample buffer for 10 min. Samples were then subjected to analysis by western blot. Primer sequences are listed in *Supplementary file 1e*.

### Histological and immunochemical analysis

The white adipose tissue was fixed in 4% formalin solution at 4 °C for 24 hr, embedded in paraffin, and sectioned at 5 µm. Deparaffinized and rehydrated sections were stained with hematoxylin and eosin (Sigma), or with reagents for Sirius red staining, or immunohistological staining of HCS LipidTOX red neutral stain (#H34467, Invitrogen) and F4/80 (#GB113373, Servicebio, china). The slides were analyzed using a confocal laser scanning microscope (CLSM, Carl Zeiss LSM800) at ×20 magnification.

### Transmission electron microscopy

For transmission electron microscopy, mouse epiWAT was dissected, sliced into small fragments of 1–2 mm each, and then fixed in 5% glutaraldehyde for 2 days. Specimens were post-fixed in 1% osmium tetroxide. After staining with 2% aqueous uranyl acetate for 2 h, the samples were sequentially dehydrated in 50%, 70%, 90%, and 100% ethanol for 15 min respectively, then embedded in epoxy resin. Ultrathin sections were cut with an EM UC7 ultramicrotome (Leica) and poststained with lead nitrate. Ultrathin sections were mounted in formvar-coated nickel grids and observed under an FEI Tecnai G2 Electron Microscope (FEI Tecnai G2).

### EdU labeling

Cell proliferation was detected with BeyoClick EdU Alexa Fluor 488 Imaging Kit (Beyotine, China). Briefly, $1×10^5$ macrophage cells or $1×10^4$ RAW264.7 cells were plated on 24-well plates. After co-cultured with 3T3-L1 cells or SVF cells, cells were gently washed twice with PBS, and further incubated with 10 µM EdU for 4 hr. Treated cells were fixed in 4% paraformaldehyde solution at room temperature for 15 min and EdU detection was carried out according to manufacturer's instructions.

### Sample size and replication

Sample size varied between experiments, depending on the number of mice allocated for each experiment. The minimum sample size was three.

### Data inclusion/exclusion criteria

All patients enrolled in this study were obese (BMI >25 kg/m²). The negative controls were normal-weight individuals (20 kg/m²≤BMI ≤ 25 kg/m²). Data or samples were not excluded from analysis for other reasons.

### Randomization

Mice used for the experiments were randomly selected and randomly assigned to experimental groups.

### Blinding

During experimentation and data acquisition, blinding was not applied to ensure tractability.

## Statistical analysis

All in vivo experiments represent individual mice as biological replicates, the exact values of *n* are reported in figure legends. The in vitro cell assay was performed in triplicates, each experiment was independently replicated at least three times. Data are presented as mean ± SEM. Comparisons were performed using the Student's *t* test between two groups or ANOVA in multiple groups. Dunn's multiple comparisons for one-way ANOVA and Fisher's least significant difference (LSD) for two-way ANOVA were used. The level of significance was set at $*p<0.05$, $**p<0.01$, $***p<0.001$. Graphpad prism 8 (GraphPad, San Diego, CA, USA) was used for all calculations.

## Acknowledgements

This work was supported by the National Natural Science Foundation of China: Grant No. 82100858, 82370804 (To FF Z), 82373925, 82070801 (To L J), 82073227 (To Y P). Supported by Natural Science Foundation of Jiangsu Province, BK20221520 (To L J), BK20200569 (To FF Z). Supported by grants from the '111' project, B16046 (To L J). Supported by the Priority Academic Program Development of Jiangsu Higher Education Institutions, PAPD (To L J), 2632023TD03 (to FF Z). Supported by China Postdoctoral Science Foundation, 2022T150726, Supported by the Fundamental Research Funds for the Central Universities 2020M671661 (To FF Z) and 2632023GR07 (To Y Y). Supported by Jiangsu Province Research Founding for Postdoctoral, 1412000016 (To FF Z). Supported by Jiangsu Province Outstanding Postdoctoral Program (2023ZB342 to Y Y). We would like to thank Xiaonan Ma for providing technical assistance of Carl Zeiss LSM 800 on the Public Experimental Platform of China Pharmaceutical University. We thank Yumeng Shen (Public Platform of State Key Laboratory of Natural Medicines, China Pharmaceutical University) for her assistance with flow analysis. We thank LetPub (https://www.letpub.com) for its linguistic assistance during the preparation of this manuscript.

## Additional information

### Funding

| Funder | Grant reference number | Author |
|---|---|---|
| National Natural Science Foundation of China | 82100858 | Fangfang Zhang |
| National Natural Science Foundation of China | 82373925 | Liang Jin |
| National Natural Science Foundation of China | 82073227 | Yi Pan |
| Natural Science Foundation of Jiangsu Province | BK20221520 | Liang Jin |
| Natural Science Foundation of Jiangsu Province | BK20200569 | Fangfang Zhang |
| National Natural Science Foundation of China | 82370804 | Fangfang Zhang |
| National Natural Science Foundation of China | 82070801 | Liang Jin |
| Jiangsu Province Outstanding Postdoctoral Program | 2023ZB342 | Yue Yang |
| Jiangsu Province Research Founding for Postdoctoral | 1412000016 | Fangfang Zhang |
| '111' project | B16046 | Liang Jin |

| Funder | Grant reference number | Author |
|---|---|---|
| Priority Academic Program Development of Jiangsu Higher Education Institutions | 2632023TD03 | Fangfang Zhang |
| Priority Academic Program Development of Jiangsu Higher Education Institutions | PAPD | Liang Jin |
| China Postdoctoral Science Foundation | 2022T150726 | Fangfang Zhang |
| Fundamental Research Funds for the Central Universities | 2020M671661 | Fangfang Zhang |
| Fundamental Research Funds for the Central Universities | 2632023GR07 | Yue Yang |

The funders had no role in study design, data collection and interpretation, or the decision to submit the work for publication.

## Author contributions

Yue Yang, Bin Huang, Data curation, Formal analysis, Investigation; Yimeng Qin, Data curation, Investigation; Danwei Wang, Investigation, Methodology; Yinuo Jin, Yi Pan, Software; Linmin Su, Yanfeng Zhang, Yumeng Shen, Visualization; Qingxin Wang, Software, Formal analysis; Wenjun Hu, Methodology; Zhengyu Cao, Writing – original draft, Project administration; Liang Jin, Funding acquisition, Writing – original draft, Writing – review and editing; Fangfang Zhang, Formal analysis, Funding acquisition, Writing – original draft, Project administration, Writing – review and editing

## Author ORCIDs

Liang Jin ⬡ https://orcid.org/0000-0002-4995-3553
Fangfang Zhang ⬡ https://orcid.org/0000-0002-1954-4345

## Ethics

All human subjects provided informed consent. All human studies were conducted according to the principles of the Declaration of Helsinki and were approved by the Ethics Committees of the Department Sir Run Run Hospital (Nanjing, China, 2023-SR-046).

Care of all animals was within institutional animal-care committee guidelines, and all procedures were approved by the animal ethics committee of China Pharmaceutical University (Permit Number: 2162326) and were in accordance with the international laws and policies (EEC Council Directive 86/609,1987).

## Decision letter and Author response

Decision letter https://doi.org/10.7554/eLife.99162.sa1
Author response https://doi.org/10.7554/eLife.99162.sa2

# Additional files

## Supplementary files

• Supplementary file 1. The supplementary tables in the manuscript. **(a)** Clinical characteristics of the patients with obese patients and normal individuals. **(b)** RNA islolated from epiWAT of wide type mice and *Mir802* KI mice, this table shows significantly changed mRNA (Log2 (FPKM (*Mir802* KI/WT))≥1). **(c)** Primer sequences used for RT-PCR. **(d)** Oligo sequences used for shRNA. € The primers used in Real-time PCR (5'–3').

• MDAR checklist

• Source data 1. Raw data of histograms.

• Source data 2. Quantification of the western blot analysis.

## Data availability

Sequencing data have been deposited in the NCBI's Sequence Read Archive (SRA) database (PRJNA1021754).

The following dataset was generated:

| Author(s) | Year | Dataset title | Dataset URL | Database and Identifier |
|---|---|---|---|---|
| Zhang F | 2024 | RNA-sequ raw data | https://www.ncbi.nlm.nih.gov/bioproject/?term=PRJNA1021754 | NCBI BioProject, PRJNA1021754 |

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
