## [Editor Report]

This important study utilizes a comprehensive array of animal and cellular models, alongside various techniques, to elucidate the mechanism by which adipose tissue miR-802 contributes to inflammation and metabolic dysfunction in obesity. The data are compelling, with consistent findings across replicates and different models. The work will be of interest to medical biologists working on obesity and metabolic diseases.

---

## [Decision Letter]

**Decision letter after peer review:**

Thank you for submitting your article "Adipocyte microRNA-802 promotes adipose tissue inflammation and insulin resistance by modulating macrophages in obesity" for consideration by *eLife*. Your article has been now reviewed by 3 peer reviewers, one of whom is a member of our Bard of Reviewing Editors, and the evaluation has been overseen by Satyajit Rath as the Senior Editor.

The reviewers have discussed their reviews with one another, and I have drafted this to help you prepare a revised submission.

Essential revisions:

1) According to the Reviewers, a deeper investigation of the relationship between fat accumulation and inflammation is missing. In other words, how do changes in fat accumulation drive changes in inflammation and metabolic function? This should be addressed in a revised version of the manuscript with additional data.

2) Reviewers required more data to demonstrate what mediates the communication between adipocytes and macrophages. This is yet another key element of the manuscript which should be addressed in a revised version.

3) Reviewers raised important questions regarding the mechanism underlying macrophage inflammatory activation upon miR-802 induction in adipocytes. The authors should address these points in part by expanding the characterization of macrophages in their models.

4) Important controls are missing and should be included in a revised version of the manuscript.

5) In addition to addressing these major points, we would expect a point-by-point response to all Reviewers' concerns.

*Reviewer #1 (Recommendations for the authors):*

In addition to the points raised in the public reviews, there are other issues that should be addressed by the authors:

1) Line 93: "obesity mice" should be "obese mice".

2) Line 95: ".we…" should be ". We…" with a capital letter.

3) Line 138: "KI" – define abbreviations the first time they appear.

4) Lines 150-151: "No change of CLSs was between two groups fed with NCD" – revise sentence.

5) Line 179: "miR-802 is required for the recruitment…" – I would change "required" for "sufficient".

6) Figure 2 and potentially others: watch for overlap between panels. Legends from panels D and O are cut.

7) Line 284: what do you mean by "various obese mice"?

8) Figure 5 and potentially others: maintain the same color code. It is confusing when color code changes across experimental groups.

9) Line 456: the word "while" should be deleted.

10) Supplementary Figure 3N: the picture is not very informative.

11) Supplementary Figure 3O: I believe "kidney" is misspelled.

12) Quantification of the blots should be presented, as well as the number of biological replicates for all blots, particularly the ones where only a representative sample is presented (e.g., Figure 5F, Figure 6 A, C, E, and G, and Figure 7B, D, H, and I).

*Reviewer #2 (Recommendations for the authors):*

Yang et al. investigated the effects of obesity on miR-802 expression and the role of miR-802 in obesity-related inflammation. The authors found that miR-802 levels are increased in insulin-resistant AT. Then, by using genetic approaches of gain- and loss-of-function, it was found that miR-802 manipulation effectively controls adipose tissue inflammation and macrophage infiltration. The authors went further to demonstrate this regulation of AT inflammation occurs by interaction of miR-802 with intracellular TRAF3, controlling NF-κB pathway activation. Overall, the study is well-written, and the techniques applied allow robust conclusions regarding the role of miR-802 in adipose tissue inflammation. I have a few suggestions for the authors to strengthen the quality of the study which I hope you find helpful.

1) In the co-culture experiments, what is the effect of co-culture on migration and invasion? Showing migration and invasion without any co-culture is an important control and would be informative to know what are the effects of a normal adipocyte secretion on macrophage function.

2) It is now recognized that adipocytes produce and release miRNAs to act in distant organs (PMID: 28492253; 36070680; 35387487). It is important to measure miR-802 in macrophages following co-culture experiments to rule out a direct effect of adipocyte-derived miR-802 in macrophages.

3) Could the authors provide the source and codes of the diets used in this study?

4) Palmitate 0.5mM used to induce insulin resistance can cause cell. Could the authors provide evidence of the absence of cell death in this treatment?

5) Although the regulation of Srebp1 by miR-802 was convincingly demonstrated, the relationship between lipid droplet formation and the engulfment of lipid droplets on macrophage phenotype is not clear. Could the authors include this limitation and open questions in the Discussion section, please?

6) As far as I understood, the BMI of lean participants ranged from 20 to 25 kg/m2. Could the authors reword this sentence in the methods section to make it clearer?

7) Why were lean and insulin-sensitive participants excluded from the correlation analysis in Figure 1J and 1K, respectively?

*Reviewer #3 (Recommendations for the authors):*

1) CD86+CD206- cannot be considered as a marker of inflammatory macrophages without an intracellular cytokine profile and CD11c expression.

2) Because miR-802 deletion affected adiposity, it is difficult to differentiate if the anti-inflammatory and pro-metabolic effects of miR-802 are due to a direct action on macrophages of other unrelated mechanisms.

3) In Figure 3 there is no pre-gate on CD11b/F4/80/CD45 cells, thus, these cells can be any type of stromal vascular cell. Also, there is no gating strategy with macrophage and viability plots.

4) There are no insights on why obesity triggers the accumulation of miR-802.

5) Raw cells do not resemble primary macrophages and conclusions drawn with this cell line cannot be expanded unless also show at least in bone marrow derived macrophages.

6) Description of Figure 4 in the text should be fully revised. It is kind of confusing.

7) CCL2/MCP1 are well known chemokines for monocytes in adipose tissue, blocking assays could help determine her effect in miR-802 overexpression.

8) Why in Figure 4H are different cell numbers in the flow plots? A dye for viability should be added in all experiments.

9) Traf3 is associated protein related to TNFR, CD40, among other receptors signaling pathways, leading, for example, to NFKB activation. However, here the authors show that increased expression of Traf3 results in reduced macrophage count and adipose inflammation and improved systemic glucose homeostasis. Also, the data appears to show the opposite result. This is confusing. There are data showing that Traf3 may also attenuate NFKB signaling, this is an interesting effect that is partially investigated in the manuscript. Overall, for this topic, it is difficult to follow the data and addressing this would improve the manuscript and the potential mechanism involved.

10) The data related to LD is confusing, the transcriptomics was performed in whole tissue and confirmed in 3T3 cells and expanded to macrophages as they were the same cell types.

11) miR-802 effects on Traf3 are studies on adipocytes, but what is the role of Traf3 in macrophages, are responsive? What does the expression of miR-802 in adipocytes trigger in macrophages?

12) Figure 6B, the Teaf3 blot appears to come from a different gel, please upload the complete gels.

13) The authors mention sometimes in the text macrophage infiltration. However, as there are no circulating macs, they probably mean monocyte infiltration.

14) Figure S4E – the SSC FSC plot is gating lymphoid cells, the population located next to this is the myeloid cell population. This may account for the high proliferation observed, which is not common in macrophages.

[Editors' note: further revisions were suggested prior to acceptance, as described below.]

Thank you for resubmitting your work entitled "Adipocyte microRNA-802 promotes adipose tissue inflammation and insulin resistance by modulating macrophages in obesity" for further consideration by *eLife*. Your revised article has been evaluated by Satyajit Rath (Senior Editor) and Marcelo Mori as the Reviewing Editor.

The manuscript has been improved but there are some remaining issues that need to be addressed, as outlined below:

While the other reviewers were satisfied with the revised manuscript, Reviewer #2 still has a remaining concern. Additionally, the manuscript would benefit from a thorough grammatical review, as there are still some errors in the text.

*Reviewer #2 (Recommendations for the authors):*

I have now read the revised the work of Yang et al. with great interest. The original manuscript was of high quality and had important findings for not just the obesity field but also microRNAs and tissue-communication fields. The authors answered almost all my questions, which I greatly appreciate. However, one point that was not completely addressed was the migration/invasion assay in Figure 4 and Supplementary Figure 4. In Supplementary Figure 4, the authors show the control (i.e. no adipocyte) for macrophage migration/invasion, and then the co-culture of adipocytes overexpressing miR-802 and Mir802 & NOX-E36.

On the other hand, Figure 4 shows migration/invasion for all co-culture but not without co-culture, which the authors infer based on the previous figure (Supplementary Figure 4) is quite low without adipocytes being co-cultured. Therefore, the direct effect of adipocyte co-culture on macrophage migration/invasion has not been shown yet. I suggest showing the migration/invasion assay, at least once, with all conditions (i.e. no adipocyte, co-culture with adipocyte with no miR-802 manipulation, and then the co-culture with miR-802 manipulation or drugs). This approach will be very informative to the reader in understanding the full context of macrophage migration/invasion behavior in the presence of adipocytes and modulation of miR-802 expression. Apart from this, I have no further suggestions other than to congratulate the authors for the nice piece of work.

---

## [Author Response]

Essential revisions:1) According to the Reviewers, a deeper investigation of the relationship between fat accumulation and inflammation is missing. In other words, how do changes in fat accumulation drive changes in inflammation and metabolic function? This should be addressed in a revised version of the manuscript with additional data.

Thanks for your good suggestions. We have addressed additional data in the revised manuscript to determine the relationship between fat accumulation and inflammation.

In our study, we have determined that overexpression *Mir802* activates the canonical and canonical NF-κB pathways, which can increase CCL2 secretion leading to recruit macrophage. Then, we clarified that *Mir802* indirectly stimulates lipogenic gene sterol regulatory element-binding protein 1 a (SREBP1a) expression via western blot (Figure 7B-D), Chip-qPCR (Figure 7F-G), DNA pull-down assays (Figure 7 H-I) and luciferase assays (Figure 7J).

As previously described, Srebp1a is a well-established regulator of lipid synthesis and required for macrophage phagocytosis lipid synthesis (doi: 10.1073/pnas.1813458115, doi: 10.1016/j.tcb.2020.09.006). Accordingly, *Mir802* overexpression significantly increased lipid synthesis (Figure 7—figure supplement 1G-H). Moreover, we found overexpression-*Mir802* 3T3-L1 cells promoted RAW264.7 cells to engulf more lipid droplets through direct contact co-culture with mature 3T3-L1 and RAW264.7 cells, while the content of lipid droplets is decreased in the knockdown-*Mir802* 3T3-L1 cells (Figure 7K). Furthermore, we also observed that adipose tissue macrophages (ATMs) of the *Mir802* KI mice could engulf more lipid droplets (Figure 7L, M). Altogether, these data show that *Mir802* indirectly regulates lipid droplet formation through SREBP1 and ultimately promotes macrophage to engulf more lipid droplets, promoting the polarization of macrophages.

2) Reviewers required more data to demonstrate what mediates the communication between adipocytes and macrophages. This is yet another key element of the manuscript which should be addressed in a revised version.

Thank you! The suggestion you provided is greatly appreciated. We have added more data to demonstrate the exact mediator regulating the communication between adipocytes and macrophages.

In the previous study, we have overexpressed or knockdown *Mir802* in the 3T3-L1 cells, but we detected almost no alterations of IL-6, IL-1β, and TNF-α levels, but the level of CCL2 was significantly higher when we cultured *Mir802*-overexpressing 3T3-L1 adipocytes alone (Figure 4I, A below). Moreover, we found without any co-culture, macrophage has no ability to migration and invasion (B)**.**

Additionally, according to the reviewer’s suggestion, emapticap pegol (also known as NOX-E36) was employed to inhibit CCL2, and the results demonstrated that blocking CCL2 in *Mir802*-overexpressed 3T3-L1 cells exhibited reduced macrophage recruitment ability. (Figure 4—figure supplement 1J).

Moreover, in our previous study, clodronate-conjugated liposomes (CLOD-liposomes) were injected into both adipose-specific *Mir802* KI mice and WT littermates via intraperitoneally to deplete macrophages. The clodronate liposomes treatment ameliorated systematic inflammation, as displayed by the decreased levels of serum inflammatory factors (TNF-α, IL-6, IL-1β, and CCL2) (Figure 4K). Remarkably, the differences in these circulating factors except CCL2 between *Mir802* KI mice and WT littermates became indistinguishable after macrophage depletion, suggesting that the macrophages are not the main source of CCL2. Furthermore, macrophage depletion also led to recover hyperinsulinemia (Figure 4L) and HOMA-IR (Supplementary Figure 4K) in *Mir802* KI mice after the treatment. Similarly, macrophage depletion with CLOD-liposomes obviously alleviated the HFD-induced glucose intolerance in *Mir802* KI mice, and abrogated the differences of these metabolic indicators between *Mir802* KI mice and WT mice (Figure 4M). Taken together, these data support macrophages as an important mediator for adipose *Mir802*–induced systemic inflammation in obesity. We have added these data in the manuscript, these data could demonstrate that *Mir802* increased CCL2 level, which is the mediates to regulate the communication between adipocytes and macrophages.

3) Reviewers raised important questions regarding the mechanism underlying macrophage inflammatory activation upon miR-802 induction in adipocytes. The authors should address these points in part by expanding the characterization of macrophages in their models.

Thank you for this important suggestion. We have added enough data to address this point.

As suggested, we have detected *Mir802* expression in the macrophage in the co-culture experiments. qRT-PCR results showed that *Mir802* expression has no difference, these results indicated that *Mir802* does not direct effect macrophage. And we have provided more additional data, the results showed that adipose *Mir802* activated NF-κB pathway, promoting adipose secreted CCL2, recruitment macrophage, and finally increasing fat inflammation.

4) Important controls are missing and should be included in a revised version of the manuscript.

Thank you for this important point. We have added important controls in the revised version of the manuscript (Figure 1G and 1H).

5) In addition to addressing these major points, we would expect a point-by-point response to all Reviewers' concerns.

Thank you for your good advice. We have responded all reviewers’ concerns point by point.

Reviewer #1 (Recommendations for the authors):In addition to the points raised in the public reviews, there are other issues that should be addressed by the authors:1) Line 93: "obesity mice" should be "obese mice".

Thank you for your good advice. We have responded all reviewers’ concerns point by point.

2) Line 95: ".we…" should be ". We…" with a capital letter.

Thank you for this insightful comment. Line 95: “.we…”have been rewritten “. We….”.

3) Line 138: "KI" – define abbreviations the first time they appear.

Thank you for this insightful comment and we apologize for the ambiguous formulation. We have defined abbreviations of KI (Knock-in) in line138.

4) Lines 150-151: "No change of CLSs was between two groups fed with NCD" – revise sentence.

Thank you for this critical suggestion. We have revised this sentence.

“No significant differences were observed in CLSs between the control and *Mir802* KI mouse groups treatment with normal chow diet (NCD, Figure 2—figure supplement 1I).”

5) Line 179: "miR-802 is required for the recruitment…" – I would change "required" for "sufficient".

Thank you for this critical suggestion. As suggested, we have changed “required” to “sufficient”. This modification will enable us to express our conclusion more accurately.

6) Figure 2 and potentially others: watch for overlap between panels. Legends from panels D and O are cut.

Thank you for this important advice and we apologize for our lack of attention to detail. We have modified it.

7) Line 284: what do you mean by "various obese mice"?

Thank you for your careful reviewing and we apologize for the ambiguous formulation. “various obese mice (Figure 5B, C and Figure 5—figure supplement 1B, C) mean HFD, *Lep^ob/ob^* and *Lepr^db/db^* mice. We have corrected it.

8) Figure 5 and potentially others: maintain the same color code. It is confusing when color code changes across experimental groups.

The encouraging comment you provided is greatly appreciated. The utilization of the same color code can indeed facilitate readers' comprehension of the manuscript. We have maintained the same color code in the figures.

9) Line 456: the word "while" should be deleted.

Thank you for your meaningful comments. According to your advice, we have deleted the while.

10) Supplementary Figure 3N: the picture is not very informative.

We appreciate your insightful comment. We have modified the Flowchart of the in vivo experiments. We hope the changes above would make you and other readers much easier to understand our manuscript.

11) Supplementary Figure 3O: I believe "kidney" is misspelled.

Thank you for your careful reviewing and we are sorry for our incorrect writing. We have corrected it.

12) Quantification of the blots should be presented, as well as the number of biological replicates for all blots, particularly the ones where only a representative sample is presented (e.g., Figure 5F, Figure 6 A, C, E, and G, and Figure 7B, D, H, and I).

Thank you for this important point. We have quantified of the blots and we have presented the number of biological replicates for Figure 5F, Figure 6 A, C, E, and G, and Figure 7B, D, H, and I. Due to the large amount of data in the manuscript, we have provided other quantification of the blots in the raw data.

Reviewer #2 (Recommendations for the authors):Yang et al. investigated the effects of obesity on miR-802 expression and the role of miR-802 in obesity-related inflammation. The authors found that miR-802 levels are increased in insulin-resistant AT. Then, by using genetic approaches of gain- and loss-of-function, it was found that miR-802 manipulation effectively controls adipose tissue inflammation and macrophage infiltration. The authors went further to demonstrate this regulation of AT inflammation occurs by interaction of miR-802 with intracellular TRAF3, controlling NF-κB pathway activation. Overall, the study is well-written, and the techniques applied allow robust conclusions regarding the role of miR-802 in adipose tissue inflammation. I have a few suggestions for the authors to strengthen the quality of the study which I hope you find helpful.1) In the co-culture experiments, what is the effect of co-culture on migration and invasion? Showing migration and invasion without any co-culture is an important control and would be informative to know what are the effects of a normal adipocyte secretion on macrophage function.

Thanks for your encouraging comments. In the previous study, we have overexpressed or knockdown *Mir802* in the 3T3-L1 cells, but we detected no alterations of IL-6, IL-1β, and TNF-α levels, but the level of CCL2 was significantly higher when we cultured *Mir802*-overexpressing 3T3-L1 adipocytes alone (Figure 4I, A below). Moreover, we found without any co-culture, macrophage has no ability to migration and invasion (B)**.**

Instead, we co-culture 3T3-L1 transfected with *Mir802* mimics or *Mir802* inhibitor and 264.7 cells (Figure 4—figure supplement 1C). We found *Mir802*-overexpressing 3T3-L1 cells promoted the migration and invasion of RAW 264.7 cells, whereas 3T3-L1 cells knocked down by anti-*Mir802* had the opposite effect (Figure 4I). We also found higher level of CCL2 in the medium conditioned with *Mir802*-overexpressed 3T3-L1 cells (Figure 4—figure supplement 1H). Additionally, emapticap pegol (also known as NOX-E36) was employed to inhibit CCL2, and the results demonstrated that blocking CCL2 in *Mir802*-overexpressed 3T3-L1 cells exhibited reduced macrophage recruitment ability. (Figure 4—figure supplement 1J).

Overall, our findings provide support for the hypothesis that obese adipose tissue upregulates *Mir802*, which facilitate adipose cells to secrete CCL2, leading to the recruitment and activation of macrophages by *Mir802*-overexpressing adipocytes.

2) It is now recognized that adipocytes produce and release miRNAs to act in distant organs (PMID: 28492253; 36070680; 35387487). It is important to measure miR-802 in macrophages following co-culture experiments to rule out a direct effect of adipocyte-derived miR-802 in macrophages.

Thank you for this insightful comment. Adipocytes did produce and release miRNAs to act in distant organs based on previously reports. In the previously study, we have detected *Mir802* expression in the macrophage in the co-culture experiments. qRT-PCR results showed that *Mir802* expression has no different, these results indicated that *Mir802* is not direct effect macrophage.

3) Could the authors provide the source and codes of the diets used in this study?

Thank you for your careful reviewing and we apologize for the ambiguous formulation. As suggested, we have added the detailed composition of high-fat diet and normal chow diet given to the mice.

4) Palmitate 0.5mM used to induce insulin resistance can cause cell. Could the authors provide evidence of the absence of cell death in this treatment?

Thank you for your kind advice. In the previously study, we used flow cytometry to test the cell death, the results showed that insulin resistance model used 0.5 mM palmitate acid almost has no effect on cell death (below). Moreover, other study also used palmitate acid to induce insulin resistance (Doi: 0013-7227/97/$03.00/0).

5) Although the regulation of Srebp1 by miR-802 was convincingly demonstrated, the relationship between lipid droplet formation and the engulfment of lipid droplets on macrophage phenotype is not clear. Could the authors include this limitation and open questions in the Discussion section, please?

We greatly appreciate your valuable input. Here, firstly, we have determined that overexpression *Mir802* activates the canonical and canonical NF-κB pathways, which can increase CCL2 secretion leading to recruit macrophage. Then, we clarified that *Mir802* indirectly stimulates lipogenic gene sterol regulatory element-binding protein 1 a (SREBP1a) expression via western blot (Figure 7B-D), Chip-qPCR (Figure 7F-G), DNA pull-down assays (Figure 7 H-I) and luciferase assays (Figure 7J).

As previously described, Srebp1a is a well-established regulator of lipid synthesis and required for macrophage phagocytosis lipid synthesis (doi: 10.1073/pnas.1813458115, doi: 10.1016/j.tcb.2020.09.006). Accordingly, *Mir802* overexpression significantly increased lipid synthesis (Figure 7—figure supplement 1G-H) in the adipose cells, and *Mir802* has no effect on the lipid synthesis of macrophage. However, we found overexpression-*Mir802* 3T3-L1 cells promoted RAW264.7 cells to engulf more lipid droplets through direct contact co-culture with mature 3T3-L1 and RAW264.7 cells, while the content of lipid droplets is decreased in the knockdown-*Mir802* 3T3-L1 cells (Figure 7K). Moreover, we also observed that adipose tissue macrophages (ATMs) of the *Mir802* KI mice could engulf more lipid droplets (Figure 7L-M). Altogether, these data show that *Mir802* indirectly regulates lipid droplet formation through SREBP1 and ultimately promotes macrophage to engulf more lipid droplets.

Jae-Ho Lee determined that Srebp1a provides lipids essential for maintaining association between the actin cytoskeletal network and plasma membranes, thus enhancing phagocytosis (doi: 10.1073/pnas.1813458115), whether *Mir802* promote macrophage to engulf lipid droplets via regulation the actin cytoskeletal network and plasma membranes still need to further research. As suggested, we have added this limitation and open questions in the Discussion section. We anticipate that these modifications will enhance the comprehensibility of our manuscript for both yourself and other readers.

6) As far as I understood, the BMI of lean participants ranged from 20 to 25 kg/m2. Could the authors reword this sentence in the methods section to make it clearer?

Thank you for this important point. We have reworded this sentence in the methods section.

7) Why were lean and insulin-sensitive participants excluded from the correlation analysis in Figure 1J and 1K, respectively?

Thank you for this critical suggestion. In the previously study we have analyzed the correlation both obese patients and normal individuals. The results showed that the BMI and HOMA-IR were positively associated with *Mir802* abundance in subcutaneous fat. At first, we want to emphasize the *Mir802* expression in the obese people, we just analyzed the obese individuals. According to your suggestion, we have added correlation analysis of the two group.

Reviewer #3 (Recommendations for the authors):1) CD86+CD206- cannot be considered as a marker of inflammatory macrophages without an intracellular cytokine profile and CD11c expression.

The suggestions you provided are greatly appreciated. In our study, we used CD86^+^CD206^-^ to identify M1 macrophages, and we also tested the M1 macrophage–related genes (*Ccl2*, *Il1b*, *Il6*, *Tnfa*, *Inos*, and *Ifn-γ*) expression via qRT-PCR, as well as measured several inflammatory factors (CCL2, IL-1β, IL-6 and TNF-α) to investigate *Mir802* regulated inflammatory. since we performed FCM, qPCR and ELISA assays to test inflammatory macrophages, these results enough to enhance our conclusion that *Mir802* intensify adipose inflammation. Moreover, our research strategy is consistent with various other reports (doi.org/10.1172/JCI123069. doi:10.1038/s41590-018-0113-3.).

Perhaps a more precise method for detecting macrophage inflammation is by co-staining CD11c with an intracellular cytokine profile, such as IL-10. While it will take a considerable amount of time to breed enough *Mir802* KI and *Mir802* KO mice for the purpose of repeating FCM assays, we should resubmit revisions within the given timeframe. Additionally, we are confident that even if we conducted FCM using CD11c and IL-10, it would not alter our conclusions.

We greatly appreciate your valuable suggestions, and we will take them into consideration for our future research.

2) Because miR-802 deletion affected adiposity, it is difficult to differentiate if the anti-inflammatory and pro-metabolic effects of miR-802 are due to a direct action on macrophages of other unrelated mechanisms.

Thanks for your encouraging comments. In the previous study, we have overexpressed or knockdown *Mir802* in the 3T3-L1 cells, but we detected no alterations of Il6, Il1b, and Tnfa levels, but the level of CCL2 was significantly higher when we cultured *Mir802*-overexpressing 3T3-L1 adipocytes alone (Figure 4I, A below). Moreover, we found without any co-culture, macrophage has no ability to migration and invasion (B)**.**

However, we found *Mir802*-overexpressing 3T3-L1 cells promoted the migration and invasion of RAW 264.7 cells via co-culture experiment (Figure 4G). We also found higher level of CCL2 in the medium conditioned with *Mir802*-overexpressed 3T3-L1 cells (Figure 4—figure supplement 1H). Additionally, emapticap pegol (also known as NOX-E36) was employed to inhibit CCL2, and the results demonstrated that blocking CCL2 in *Mir802*-overexpressed 3T3-L1 cells exhibited reduced macrophage recruitment ability. (Figure 4—figure supplement 1J).

Moreover, in our previous study, clodronate-conjugated liposomes (CLOD-liposomes) were injected into both adipose-specific *Mir802* KI mice and WT littermates via intraperitoneally to deplete macrophages. The clodronate liposomes treatment ameliorated systematic inflammation, as displayed by the decreased levels of serum inflammatory factors (Tnfα, Il6, Il1b, and CCL2) (Figure 4K). Remarkably, the differences in these circulating factors except CCL2 between *Mir802* KI mice and WT littermates became indistinguishable after macrophage depletion, suggesting that the macrophages are not the main source of CCL2. Furthermore, macrophage depletion also led to recover hyperinsulinemia (Figure 4L) and HOM-IR (Supplementary Figure 4G) in *Mir802* KI mice after the treatment. Similarly, macrophage depletion with CLOD-liposomes obviously alleviated the HFD-induced glucose intolerance in *Mir802* KI mice, and abrogated the differences of these metabolic indicators between *Mir802* KI mice and WT mice (Figure 4M). Taken together, these data support macrophages as an important mediator for adipose *Mir802*–induced systemic inflammation in obesity. We have added these data in the manuscript, these data could enhance our results that anti-inflammatory and pro-metabolic effects of *Mir802* are due to a direct action on macrophages.

3) In Figure 3 there is no pre-gate on CD11b/F4/80/CD45 cells, thus, these cells can be any type of stromal vascular cell. Also, there is no gating strategy with macrophage and viability plots.

Thanks for your good suggestions. Firstly, in our study, we have used Zombie NIR Fixable Viability Kit (biolegend) to stain SVF, which can exclude dead cells. Macrophage were identified as CD11b^+^F4/80^+^. We are sorry for our inaccurate expression. Now, detail protocol has been added in the methos of revision manuscript, and we have added all pre-gate of F4/80/CD11b in the Figure. We hope this modification can make you and other readers understanding our manuscript more easily.

4) There are no insights on why obesity triggers the accumulation of miR-802.

We express our gratitude for your invaluable suggestion. According to our previous study, in the islet, we have determined that *FoxO1* could bind to *Mir802* promoter, and increased *Mir802* expression (*Nature communications* 2020). *FoxO1* is predominantly expressed in WAT and is upregulated in the WAT of the obese mice (doi: 10.1016/s1534-5807(02)00401-x, doi: 10.3389/fendo.2023.1286838). Here, we also found that *FoxO1* can bind to *Mir802* promoter in the 3T3-L1 cells via ChIP assays, and over-expression *FoxO1* upregulated *Mir802* expression. We have added these data in the manuscript, we hope this change can enhance our conclusion.

5) Raw cells do not resemble primary macrophages and conclusions drawn with this cell line cannot be expanded unless also show at least in bone marrow derived macrophages.

Thank you for your careful reviewing. We apologize for our lack of foresight, though many studies used resident peritoneal macrophage (doi.org/10.1016/j.ymthe.2021.03.013, doi.org/10.1080/15548627.2021.1985338), while as you pointed the function of peritoneal macrophages was lower than bone marrow derived macrophages (BMDMS). Thus, according to your suggestion, we have repeated our experiments in the BMDMS. Both in peritoneal macrophages and BMDMS, we obtained the same conclusions that obesity-SVF cells secreted chemokines, improving adipose cells to recruit macrophages.

6) Description of Figure 4 in the text should be fully revised. It is kind of confusing.

Thank you for your careful reviewing. The purpose of Figure 4 is to demonstrate that *Mir802* does not directly affect macrophage function, while it enhances adipose cells to the secretion of chemokines, particularly CCL2, regulating the communication between adipose cells and macrophages.

According to your good advice, we have added some important experiment to enhance our conclusion. Firstly, we detected *Mir802* levels in the macrophages after co-cultured with adipose cells, and qPCR results showed that *Mir802* levels have no change in the macrophage (Figure 4 F and H), which indicated that *Mir802* dose not directly affect macrophage. Moreover, emapticap pegol (also known as NOX-E36) was employed to inhibit CCL2, and the results demonstrated that blocking CCL2 in *Mir802-*overexpressed 3T3-L1 cells exhibited reduced macrophage recruitment ability. The results of the co-culture experiments showed that *Mir802* promoted adipose cells to secret CCL2 to recruit macrophage (Figure 4—figure supplement 1J). We rewrote this part; we hope the changes above would make you and other readers much easier to understand our manuscript.

7) CCL2/MCP1 are well known chemokines for monocytes in adipose tissue, blocking assays could help determine her effect in miR-802 overexpression.

Thank you for this important point. As suggested, we have used emapticap pegol (also known as NOX-E36) to inhibit CCL2 levels induced by *Mir802*, and the results demonstrated that blocking CCL2 in *Mir802*-overexpressed 3T3-L1 cells exhibited reduced macrophage recruitment ability (Figure 4—figure supplement 1J). these data can enhance our conclusion that *Mir802* promoted adipose cells to secret CCL2 to recruit macrophage.

8) Why in Figure 4H are different cell numbers in the flow plots? A dye for viability should be added in all experiments.

Thank you for your careful reviewing. In our study, a dye for viability was added in all experiments. We collected the RAW264.7 cells which co-cultured with 3T3-L1 cells, then we incubated anti-CD86 and anti-CD206. Next, we tested the percentage of CD86^+^CD206^-^ and CD86^-^CD206^+^ via FCM assays. We carefully checked our results, and we found the cell numbers used to analyze in the Figure 4H almost same.

9) Traf3 is associated protein related to TNFR, CD40, among other receptors signaling pathways, leading, for example, to NFKB activation. However, here the authors show that increased expression of Traf3 results in reduced macrophage count and adipose inflammation and improved systemic glucose homeostasis. Also, the data appears to show the opposite result. This is confusing. There are data showing that Traf3 may also attenuate NFKB signaling, this is an interesting effect that is partially investigated in the manuscript. Overall, for this topic, it is difficult to follow the data and addressing this would improve the manuscript and the potential mechanism involved.

Thank you for your careful reviewing. Firstly, based on previous reports, Traf3 is an inhibitory protein of NF-κB pathway. Activation of the NF-κB pathway needs degradation of TRAF3 (doi:10.1038/nature11831, doi.org/10.1074/jbc.M413634200), we observed the same phenomenon in our study. We determined that *Mir802* activated NF-κB pathway via inhibiting *Traf3* expression. In the adipose activation NF-κB pathway intensifies adipose tissue inflammation and insulin resistance (doi.org/10.2337/db22-0284), we also found that selectively ablated *Mir802* in adipose tissue inhibited NF-κB pathway reducing macrophage count and adipose inflammation and improving systemic glucose homeostasis.

10) The data related to LD is confusing, the transcriptomics was performed in whole tissue and confirmed in 3T3 cells and expanded to macrophages as they were the same cell types.

Thank you for this important point. The data related to LD, we were performed both in vivo and in vitro. Firstly, transcriptomics was performed in the epiWAT of *Mir802* KI mice and their WT littermates, the results revealed that the lipogenic gene sterol regulatory element-binding protein 1 a (SREBP1a) was increased, Srebp1a is a well-established regulator of lipid synthesis. Accordingly, we determined that *Mir802* KI mice significantly increased the number of lipid droplets, while *Mir802* KO mice significantly decreased the number of lipid droplets (Figure 7—figure supplement 1G). We also observed the same phenomenon in the 3T3-L1 cells transfected with *Mir802* mimic or *Mir802* inhibitor (Figure 7—figure supplement 1H).

3T3-L1 cells derived from the mouse fibroblast line 3T3, which can synthesis triglycerides after differentiation 3T3-L1 cells are an ideal cell model for fat metabolism (doi: 10.1038/s41467-024-45899-4, doi: 10.1016/j.cbi.2021.109538.).

11) miR-802 effects on Traf3 are studies on adipocytes, but what is the role of Traf3 in macrophages, are responsive? What does the expression of miR-802 in adipocytes trigger in macrophages?

Thank you for your meticulous review. Indeed, we have convincingly demonstrated that *Mir802* inhibited *Traf3* expression via directly targeting *Traf3*. In our previous study, we found *Mir802* was enriched in the adipocyte, and lower expressed in the SVF cells (Figure 1D). Moreover, we sorted F4/80^+^/CD11b^+^ by flow cytometry, qPCR results revealed that *Mir802* expression was very low. Furthermore, our co-culture experiments indicated that ectopic expression of *Mir802* in the 3T3-L1 cells have not changed *Mir802* expression in the macrophage (Figure 4H). Overall, *Mir802* was not directly regulated macrophage function. *Mir802* in adipocyte regulated adipocyte secretion chemokines, which in tun to affect macrophage function.

12) Figure 6B, the Teaf3 blot appears to come from a different gel, please upload the complete gels.

Thank you for this important point. All complete gels have uploaded when we submitted the manuscript. Now, we have provided the complete gels of TRAF3 of the Figure 6B. TRAF3 blots was come from the same gel.

13) The authors mention sometimes in the text macrophage infiltration. However, as there are no circulating macs, they probably mean monocyte infiltration.

Thanks for your good advice. The term "macrophage infiltration" was primarily used to describe the regulation of macrophage infiltration by *Mir802* in mice (Figure 2, Figure 3). Manuscript has been thoroughly reviewed and we have made corrections where it was deemed inappropriate.

14) Figure S4E – the SSC FSC plot is gating lymphoid cells, the population located next to this is the myeloid cell population. This may account for the high proliferation observed, which is not common in macrophages.

Thank you for your careful reviewing. We are very sorry, perhaps due to our unclear writing, causing a misunderstanding for you. The results of flow cytometer of Figure 4—figure supplement 1E was KI67 results in the cell line of RAW264.7 cells. Here, *Mir802* mimics or *Mir802* inhibitor was transfected into the mature 3T3-L1 cells, *Mir802* ectopically expressed 3T3-L1 cells on the macrophage cell line RAW264.7 in co-culture (Figure 4—figure supplement 1C). Then we investigate *Mir802* regulated RAW264.7 proliferation via EdU and FCM. We found that the *Mir802* levels have no different in the RAW264.7 cells (Figure 4H), *Mir802*-overexpressing 3T3-L1 cells had no effect on the proliferation of RAW264.7 cells (Figure 4—figure supplement 1E, F). We have rewritten it, and we hope the changes above would make you and other readers much easier to understand our manuscript.

[Editors' note: further revisions were suggested prior to acceptance, as described below.]

While the other reviewers were satisfied with the revised manuscript, Reviewer #2 still has a remaining concern. Additionally, the manuscript would benefit from a thorough grammatical review, as there are still some errors in the text.Reviewer #2 (Recommendations for the authors):I have now read the revised the work of Yang et al. with great interest. The original manuscript was of high quality and had important findings for not just the obesity field but also microRNAs and tissue-communication fields. The authors answered almost all my questions, which I greatly appreciate. However, one point that was not completely addressed was the migration/invasion assay in Figure 4 and Supplementary Figure 4. In Supplementary Figure 4, the authors show the control (i.e. no adipocyte) for macrophage migration/invasion, and then the co-culture of adipocytes overexpressing miR-802 and Mir802 & NOX-E36.On the other hand, Figure 4 shows migration/invasion for all co-culture but not without co-culture, which the authors infer based on the previous figure (Supplementary Figure 4) is quite low without adipocytes being co-cultured. Therefore, the direct effect of adipocyte co-culture on macrophage migration/invasion has not been shown yet. I suggest showing the migration/invasion assay, at least once, with all conditions (i.e. no adipocyte, co-culture with adipocyte with no miR-802 manipulation, and then the co-culture with miR-802 manipulation or drugs). This approach will be very informative to the reader in understanding the full context of macrophage migration/invasion behavior in the presence of adipocytes and modulation of miR-802 expression. Apart from this, I have no further suggestions other than to congratulate the authors for the nice piece of work.

Thanks for your encouraging comments. According to your suggestion, we have performed the migration/invasion assay with no adipocyte, co-culture with adipocyte, co-culture with adipocyte transfected *Mir802* mimics, co-culture with adipocyte transfected *Mir802* mimics and added emapticap pegol (also known as NOX-E36, CCL2 inhibitor). The results showed that no adipocyte, RAW 264.7 cells almost have no ability to migration and invasion, co-culture with *Mir802* mimics promoted the migration and invasion of RAW 264.7 cells compared to co-culture with miR-NC, while blocking CCL2 in *Mir802*-overexpressed 3T3-L1 cells exhibited reduced RAW 264.7 cells recruitment ability (Figure 4—figure supplement 2C). Overall, our findings provide support for the hypothesis that obese adipose tissue upregulates *Mir802*, which facilitate adipose cells to secrete CCL2, leading to the recruitment and activation of macrophages by *Mir802*-overexpressing adipocytes. We have added a schematic illustration in the Figure 4—figure supplement 2E. We hope these changes will be very informative to the reader in understanding the full context of macrophage migration/invasion behavior in the presence of adipocytes and modulation of *Mir802* expression.